# Comparison of Target Detectors to Identify Icebergs in Quad-Polarimetric L-Band Synthetic Aperture Radar Data

**Johnson Bailey** *[iD], **Armando Marino and Vahid Akbari**

Department of Biological and Environmental Sciences, Faculty of Natural Sciences, University of Stirling, Stirling FK9 4LA, UK; armando.marino@stir.ac.uk (A.M.); vahid.akbari@stir.ac.uk (V.A.)
* Correspondence: johnson.bailey@stir.ac.uk

**Abstract:** Icebergs represent hazards to ships and maritime activities and therefore their detection is essential. Synthetic Aperture Radar (SAR) satellites are very useful for this, due to their capability to acquire data under cloud cover and during day and night passes. In this work, we compared six state-of-the-art polarimetric target detectors to test their performance and ability to detect small-sized icebergs <120 m in four locations in Greenland. We used four single-look complex (SLC) ALOS-2 quad-polarimetric images from JAXA for quad-polarimetric detection and we compared with dual-polarimetric detectors using only the channels *HH* and *HV*. We also compared these detectors with single-polarimetric intensity channels and we tested using two scenarios: open ocean and sea ice. Our results show that the multi-look polarimetric whitening filter (MPWF) and the optimal polarimetric detector (OPD) provide the most optimal performance in quad- and dual-polarimetric mode detection. The analysis shows that, overall, quad-polarimetric detectors provide the best detection performance. When the false alarm rate ($P_F$) is fixed to $10^{-5}$, the probabilities of detection ($P_D$) are 0.99 in open ocean and 0.90 in sea ice. Dual-polarimetric or single-polarimetric detectors show an overall reduction in performance (the ROC curves show a decrease), but this degradation is not very large (<0.1) when the value of false alarms is relatively high (i.e., we are interested in bigger icebergs with a brighter backscattering >120 m, as they are easier to detect). However, the differences between quad- and dual- or single-polarimetric detectors became much more evident when the $P_F$ value was fixed to low detection probabilities $10^{-6}$ (i.e., smaller icebergs). In the single-polarimetric mode, the *HV* channel showed $P_D$ values of 0.62 for open ocean and 0.26 for sea ice, compared to values of 0.81 (open ocean) and 0.77 (sea ice) obtained with quad-polarimetric detectors.

**Keywords:** detection; icebergs; sea ice; polarimetry; SAR

## 1. Introduction

Satellite observations in polar regions have become increasingly important in the field of remote sensing. The Arctic has been warming in recent decades [1]. Maritime activities such as commercial shipping and oil extraction bring people and resources into Arctic regions [2]. Icebergs represent a climate change indicator since they provide visible evidence of temperature and precipitation changes, and they are hazards to such maritime activities and ecological disasters that may rise from collisions [3]. Smaller icebergs that have sources from Greenland glaciers are estimated to range from 10,000 to 30,000 annually [4], and therefore are significant hazards to ships. Detection ability is paramount to ensure the safety of maritime shipping [5]. Synthetic Aperture Radar (SAR) is well known for the ability to function in harsh environments with no solar illumination and it is especially useful in areas with extended cloud cover or polar nights [6]. For this reason, it has been routinely used to monitor the Arctic [7]. The detection of icebergs has seen major progress within the last ten years [8–12]. Detectors have also been proposed based on the polarimetric behaviour of targets (i.e., the structures of the scattering and covariance matrices). There are two types of detectors which have been used: those that are designed

for single-channel data (single polarimetric), and those based on multi-channel polarimetric information (dual polarimetric and quad polarimetric). Different detectors include the intensity dual-polarisation ratio anomaly detector (iDPolRAD) [13], the polarimetric notch filter (PNF) [14], the polarimetric match filter (PMF) [15], the reflection symmetry detector [16], the multi-look polarimetric whitening filter (MPWF) [17] and the optimal polarimetric detector (OPD) [18]. Most iceberg detectors apply methodologies previously developed to detect ships but new detection algorithms have recently been introduced for ice bodies. These include icebergs and sea ice. The complex nature of icebergs and sea ice unfortunately does not allow for the discrimination of icebergs with respect to other large sea ice features such as hummocks. A new detector (iDPolRaD) devised by Marino et al. [13] has tried to address the problem of detecting small icebergs (<120 m) embedded within sea ice [19].

The clean sea polarimetric characteristic at the C and L bands can be mostly modelled using a Bragg surface [20], since, in open water, specular reflection is produced without wind effects. In contrast, icebergs show different polarimetric characteristics [21]. Work carried out by Dierking et al. [21] showed that icebergs strongly exhibit volume scattering. This was echoed by work carried out by Bailey et al. [22] on icebergs in Greenland, which showed that targets exhibit a mix of volume and surface scattering, with minimal dihedral behaviour. This is likely due to the effects of these irregularly shaped icebergs toppling over, as explained in Bailey et al. [22]. As smaller irregular icebergs begin to drift away from their source, they begin to melt, affecting the centre of mass, which can lead to them toppling over. This introduces liquid water on the surface of the ice body that increases the dielectric constant and reduces the penetration. Additionally, the area that was previously in water is more saline, which also increases the dielectric constant. In return, this makes the scattering more directional. The toppling over reduces the effect of horizontal surfaces but it also increases the effect of multiple reflections and corners. Scattering behaviour can also be explained by ice volume characteristics. Several papers have reported the polarimetric scattering behaviour of other mediums such as permafrost, snow, soil and other coherent targets [23–26]. It is known that the polarimetric behaviour of a target can be affected by pixel size, and so multi-analysis is required in different window sizes [21,22]. However, detectors can still identify icebergs at sizes of approximately 1–4 pixels. Several papers [19,27,28] highlight the ability to detect icebergs of medium to large size (120 m or larger). Nevertheless, as pointed out in Soldal et al. [19], work is still required to improve the detection ability for icebergs smaller than 120 m. These icebergs are mostly calving off glacier tongues and ice sheets in the Arctic, and their smaller masses can go undetected due to factors such as the sea state and meteorological conditions [29,30].

The aim of this paper is to compare six different detectors applied to ALOS-2 PALSAR data for iceberg detection in Greenland. ALOS-2 is an L-band quad-polarimetric SAR and therefore has a high spatial resolution (4.5 × 5.1 m) and L-band wavelength (~23 cm) which can penetrate ice bodies and reveal more volume scattering. These aims will be achieved by comparing previous detection methods that have originally been applied to ship detection and extending their use with L-band data on icebergs. In order to compare detection performance, we considered two scenarios: icebergs found in open water, and icebergs embedded within sea ice. Due to volume scattering and multiple reflection, icebergs are expected to have a cross-polarisation backscattering coefficient which is higher than the surrounding sea or sea ice. This work is the first comparison of several detectors applied to L-band polarimetric SAR (PolSAR) for Arctic icebergs.

The paper is organised as follows. Detection methodologies are presented in Section 2. The data are introduced in Section 3, the methods are outlined in Section 4 and results are given in Section 5. Discussions and conclusions are outlined in Sections 6 and 7.

*Polarimetry*

The polarimetric backscattering property of a single coherent target can be characterised according to the elements of the scattering matrix [*S*]. In a quad-polarimetric (also known as full-polarimetric) system, the matrix is [6,31]:

$$[S] = \begin{bmatrix} HH & HV \\ VH & VV \end{bmatrix} \tag{1}$$

where *H* is a linear horizontal wave and *V* is a linear vertical wave. The first letter indicates the transmitted wave and the second indicates the received wave. The elements of the matrix are the complex backscattering and they are called polarisation channels. *HH* and *VV* are known as co-channels and *VH* and *HV* are known as cross-channels. A *single* target is any deterministic (fully polarised) target that does not vary spatially or temporally. It is characterised by one scattering matrix.

The matrix can also be arranged into a scattering vector.

$$\underline{k} = \frac{1}{2} Tr([S]\boldsymbol{\psi}) = [k_1, \ k_2, \ k_3, \ k_4]^T \tag{2}$$

*Tr* is the sum of the two diagonal elements of the scattering matrix. *T* refers to the matrix transpose and $\boldsymbol{\psi}$ refers to a set of $2 \times 2$ Hermitian matrices [6]. Here, the scattering vector in (2) is a four-dimensional complex vector; but in the case of a reciprocal medium and a monostatic sensor, then *HV* = *VH* and therefore the vector reduces to a three-dimensional form. In other words, where the wave is transmitted and received at approximately the same position, the scattering vector is a complex vector in three dimensions. The targets on the ground are often distributed and non-deterministic (i.e., the target has a random scattering so that the polarimetric behaviour changes spatially and temporally), and they are known as *partial* targets. In the case of icebergs, it cannot be said in advance whether they are single or partial targets, since this depends on the dimensions and distribution of iceberg scatterers [22]. Due to the statistical nature, a partial target cannot be characterised by a single scattering matrix. We can extract the second-order statistics of a partial target using a covariance matrix [*C*] or a coherency matrix [*T*] if we are expressing the target in terms of a Pauli vector:

$$[T] = \underline{k} \cdot \underline{k}^{*T} \tag{3}$$

where $\langle . \rangle$ is an averaging operator (i.e., spatially averaging over neighbour pixels) and * refers to the complex conjugate. In order to manifest the underlying physics in an easier way, we may express the components of a matrix with the Pauli vector.

$$\underline{k} = [s_{HH} + s_{VV}, \ s_{HH} - s_{VV}, \ 2s_{HV}]^T \tag{4}$$

In Bailey et al. [22], it was shown that at the L band, icebergs seem to exhibit a mix of surface scattering (which is especially strong in the $S_{HH} + S_{VV}$ channel) and a significant volume component (which appears in the $2S_{HV}$, channel). This is because dry icebergs have reflective surfaces/interfaces and volume scattering from the ice body, whereas wet icebergs have surfaces that scatter more in the specular direction that, if oriented toward the sensor, may still produce a substantial surface scattering. This is supported by Akbari et al., who found that under melting conditions, wet glaciers scatter in the specular direction [32]. Additionally, icebergs in the Arctic are not expected to be tabular as they break off glacier tongues and therefore may have very random orientation of the corners, which are expected to have a strong cross-polarisation.

## 2. Detectors

In this section, we determine and discuss various target detectors and review the literature that describes their application. Some (but not all) of these state-of-the-art

detectors have been validated only with SAR imagery of ships. This work extends the analysis to icebergs and to the use with the L band.

## 2.1. The IDPolRAD and the DPolRAD

These detectors have been designed to identify icebergs embedded within sea ice. They are based on the intensity of the cross-and co-polarisation channels. The detector therefore identifies an increase in depolarisation. The algorithm consists of two boxcar filters in two window sizes within the *HV* and *HH* intensity images.

$$\Lambda_a = \frac{\left\langle |HV|^2 \right\rangle_{\text{target}} - \left\langle |HV|^2 \right\rangle_{\text{clutter}}}{\left\langle |HH|^2 \right\rangle_{\text{clutter}}} > T_\Lambda \tag{5}$$

The two window sizes $<>_{\text{target}}$ and $<>_{\text{clutter}}$ denote spatial sample averaging within the target and clutter windows, respectively, as is the case for boxcar filters [33]. The clutter window is larger than the target window. $T_\Lambda$ is a threshold. Marino et al. [13] have provided an extensive derivation of the formula. We tested this detector for performance on the iceberg dataset.

In general, a detection is triggered if an iceberg of the right size is found in the target window, which increases the value of $\Lambda_a$. The size of the detectable iceberg is dependent on the size of the target and clutter windows. The threshold is set by using a constant false alarm rate (CA-CFAR) as in other detectors. An iceberg that is much larger than the target window will not trigger a detection. If the surface clutter is spatially homogeneous, then the numeric value of the detector is equal to zero and if there is a reduction in volume (the volume scattering reduces from the clutter to target window), the numeric value of the detector becomes negative. The detector was tested using Sentinel-1 dual-polarimetric data [13]. First, the test statistic is applied, which improves the contrast between target and clutter before applying the threshold. The detector improved the contrast (the intensity of the target over the intensity of the clutter) between icebergs and sea ice clutter by up to 75-fold, greatly increasing the probability of accurate detection. The average sea ice-clutter was found to reduce by a factor of 35. Here, the detector is tested on L-band ALOS-2 data for the very first time. Several other papers test the detector on ship detection and found similar performances better than that of the CFAR method [34]. We also try for the first time a different version of the DPolRAD, which exploits quad-polarimetric data:

$$\Lambda_b = \frac{\left\langle |HH - VV|^2 \right\rangle_{\text{target}} - \left\langle |HH - VV|^2 \right\rangle_{\text{clutter}}}{\left\langle |HH + VV|^2 \right\rangle_{\text{clutter}}} > T_\Lambda \tag{6}$$

The intensity dual-polarisation ratio anomaly detector (iDPolRAD) is similar to the DPolRAD, with the difference being that the cross-channel *HV* is multiplied as two boxcar filters are applied over the *HV* and *VV* intensity images. Here, the scattering properties between the clutter and the target are exploited. The detector can detect anomalies depending on whether the target is producing an increase in volume scattering or multiple reflections. The detector is written as follows:

$$I = \frac{\left\langle |VH|^2 \right\rangle_{\text{target}} - \left\langle |VH|^2 \right\rangle_{\text{clutter}}}{|VV|^2_{\text{clutter}}} \sigma^0_{\text{HV}} \tag{7}$$

$$\Lambda_a \sigma^0_{\text{HV}} > T_\Lambda \tag{8}$$

The data are calibrated as normalised radar cross-sections (RCS), shown by $\sigma^0$. $\Lambda$ is a term that can be written as follows:

$$\Lambda_a = \rho_{\text{ring}} \frac{1 + c}{R_\rho^{-1} + cRVH^{-1}} - \rho_{\text{clutter}} \tag{9}$$

where $\rho$ is known as the depolarisation ratio and used to determine whether the estimation is performed in the clutter window or the area between the target and clutter window (ring). *RVH*, *c* and *Rρ* are derived from the following:

$$RVH = \frac{\left\langle |VH|^2 \right\rangle_{\text{target}}}{\left\langle |VV|^2 \right\rangle_{\text{ring}}} \tag{10}$$

$$c = \frac{N_{\text{clutter}}}{N_{\text{target}}} \tag{11}$$

$$R\rho = \frac{\rho_{\text{target}}}{\rho_{\text{ring}}} \tag{12}$$

where $N_{\text{clutter}}$ and $N_{\text{target}}$ are the number of pixels inside the clutter and target windows. The detector was tested in a case study in which SAR images were acquired from the Sentinel-1 satellite.

### 2.2. The Polarimetric Notch Filter

This method is based on the fact that targets at sea and the sea clutter have different polarimetric properties. It is based on the Geometrical Perturbation analysis proposed in Marino. It considers the full-polarimetric information (phase or intensity), and assumes that the surrounding sea is homogeneous (has a similar scattering behaviour). The expression for the GP-PNF is presented here.

$$\gamma_{\text{n}} = \frac{1}{\sqrt{1 + \frac{RedR}{\underline{t}^{*T}\,\underline{t} - \left|\underline{t}^{*T}\,\underline{t}_{\text{clutter}}\right|^2}}} \tag{13}$$

$$\gamma_{\text{n}} > T_{\text{n}} \tag{14}$$

where $\underline{t}$ is the partial feature vector (a six-dimensional complex vector obtained by stacking the independent elements of the covariance matrix) for the target area, $T_n$ is the threshold and *RedR* is set as a constant for the minimum target to be detected and avoids numerical errors when computing the detection mask. The sea clutter $\underline{t}_{\text{clutter}}$ is the normalised partial feature vector for the sea background. As this algorithm is focused on targets different from the sea (not just ships), it can also be used for the detection of other polarised targets, such as icebergs. In some cases, acquiring four polarisations is not feasible, and thus a dual-polarimetric detector was proposed based on the GP-PNF. The detector expression is shown:

$$\gamma_{\text{dn}} = \frac{1}{\sqrt{1 + \frac{RedR}{t_{\text{d}}^{*T}\,t_{\text{d}} - \left|t_{\text{d}}^{*T}\,\hat{t}_{\text{dclutter}}\right|^2}}} \tag{15}$$

$$\gamma_{\text{dn}} > T_{\text{dn}} \tag{16}$$

where $\hat{t}_{\text{dclutter}}$ refers to the dual-polarised partial feature vector of the sea clutter and $T_{\text{dn}}$ is a threshold.

### 2.3. The Polarimetric Match Filter

The polarimetric match filter (PMF) was proposed by Novak et al. and optimises the contrast between targets and the clutter [15]. The difference with the GP-PNF is that in the PMF, we apply an optimisation of the contrast, whereas the PNF considers mean information. In particular, the highest contrast is achieved through selecting the specific scattering mechanism $\underline{\omega}$ that optimises this. In other words, the detector processes

the polarimetric return to provide maximum target-to-clutter ratio. The expressions are as follows:

$$\Lambda_{\mathrm{m}} = \frac{\underset{\omega \epsilon c^3}{max} \; \underline{\omega}^{*T} | C_{\text{target}} | \underline{\omega}}{\underline{\omega}^{*T} | C_{\text{clutter}} | \underline{\omega}} \tag{17}$$

$$\Lambda_{\mathrm{m}} > T_{\mathrm{m}} \tag{18}$$

$\underline{\omega}$ is defined either as a scattering mechanism or a projection vector which is not a scattering vector. $C_{\text{target}}$ and $C_{\text{clutter}}$ are general covariance matrices for the target and clutter, respectively. $T_{\mathrm{m}}$ is a threshold value. Since the quadratic forms represent power of a specific projection vector, this detector finds the optimal projection vector in the space of unitary polarimetric targets that provides the maximum ratio between the power of target over clutter. For the PMF, we present the maximum and minimum eigenvalues as sigma1 ($\sigma_1$) and sigma3 ($\sigma_3$).

### 2.4. The Reflection Symmetry Detector

The reflection symmetry detector is derived on the basis of scattering reflection symmetry and was proposed by Nunziata et al. [16]. The detection is built using the complex values of *HH* and *HV*, with a cross-correlation evaluated in the element $C_{12}$ of the covariance matrix. In theory, $C_{12}$ should be equal to zero if the scenario is reflection symmetric. From this assumption, the $C_{12}$ element should equal zero for the sea as a homogeneous surface, whereas targets such as ship and icebergs should return a higher $C_{12}$ value, since heterogeneous asymmetric scatterers compose them. The expression is as follows:

$$XC = |\langle S_{\text{HH}} S_{\text{HV}}^* \rangle| \tag{19}$$

$$XC > T_r \tag{20}$$

### 2.5. Optimal Polarimetric Detector

The optimal polarimetric detector (OPD) was proposed by Novak et al. and is based on the likelihood ratio test (LRT) under complex Gaussian statistics [15,18]. The LRT can be given when both the target and clutter distributions are known. The optimal polarimetric detector takes into account both the target to clutter ratio (TCR) and the speckle reduction.

$$z = \underline{X}^* \Sigma_{\mathrm{c}}^{-1} \underline{X} - \left( \underline{X} - \overline{X}_{\mathrm{t}} \right)^* (\Sigma_{\mathrm{t}} + \Sigma_{\mathrm{c}})^{-1} \left( \underline{X} - \overline{X}_{\mathrm{t}} \right) \tag{21}$$

$$z > T_o \tag{22}$$

Here, $\underline{X}$ and $\overline{X}_{\mathrm{t}}$ are scattering vectors and $\Sigma_{\mathrm{t}}$ and $\Sigma_{\mathrm{c}}$ are covariance matrices for target and clutter. $T_o$ is a threshold value. This function is a quadratic used to construct an image. In the case of icebergs, it is difficult to say what the covariance matrix of the radar backscatter of a general iceberg is. Therefore, in the following, we will use a scaled identity matrix assuming icebergs are fully depolarised. This is a gross approximation, which may limit the detector performance. We leave this as future work to find a better approximation of the covariance matrix of icebergs.

### 2.6. The Polarimetric Whitening Filter

The polarimetric whitening filter (PWF) is a detector that is designed to maximally reduce the speckle variation [17]. It has been shown that, in some circumstances, the PWF provides a similar performance to the OPD in PolSAR images. The PWF was also extended to multi-look scenarios [35] and this is the version we will use in this work. Performance of the PWF depends on the quality of estimating the clutter. If the estimation of the clutter covariance matrix is closer to the real one, then we achieve the best performance. The formula of the PWF is as follows.

$$z = \underline{X}^* \Sigma_{\mathrm{c}}^{-1} \underline{X} > T \tag{23}$$

$$z > T_{pwf} \tag{24}$$

where $T_{pwf}$ is a threshold. However, the PWF has been extended for use in multi-look estimation by Liu et al. [35], and Lopes and Sery [36]. The output for the MPWF is as follows:

$$z = \frac{1}{L} \sum_{i=1}^{L} y_i^* \Sigma^{-1} y_i = tr\left(r^{-1}Y\right) \tag{25}$$

$$z > T_{mpwf} \tag{26}$$

Here, $tr(.)$ is a trace operator, $y$ is the speckled image and $L$ is the number of independent samples. $T_{mpwf}$ is a threshold value. The covariance matrix for the Gaussian speckle is represented by $r$, and $Y$ is a random matrix that is only affected by speckle. The term $z$ is known to obey the Gamma distribution [37]. However, it is difficult to say whether the probability density function (pdf) of the scatterers in icebergs obeys the Gamma distribution and in future work we will compare histograms to see which pdf best fits the scatterers. Since there is an absence of texture, the covariance matrix obeys the Wishart distribution [15]. In the case of a scenario with texture, the MPWF is rewritten as:

$$z = \frac{1}{L} \sum_{i=1}^{L} k_i^* \Sigma^{-1} k_i = tr\left(\Sigma^{-1}C\right) = \frac{t}{E\{\tau\}} tr\left(r^{-1}Y\right) = \tilde{\tau}x \tag{27}$$

where $\tau$ is a unitary texture variable, $k$ is the scattering vector and $x = tr(r^{-1}Y)$. When $L = 1$, the gamma distribution is presented in the single-look complex (SLC) case. Akbari et al. [38] has also shown that L can be approximated by the Fisher (F) distribution [39]. If $L$ increases, then the MPWF can be approximated by the gamma distribution. $C$ is the multi-look covariance matrix and sigma represents the statistical mean of multi-look covariance matrix $C$. The same threshold is applied as in (26). Here, the statistics of $z$ are unknown under texture models.

### 3. Dataset

All ALOS-2 data were collected in four locations in Greenland. Areas surveyed were in close proximity to named Greenlandic glaciers; data for which are available [40]. These glaciers are the Hammer glacier, Nakkala glacier, Apuseeq glacier and Morell glacier and all of them are ocean terminating. This meant that the abundance of icebergs was high. Two scenarios were considered. First, two locations in which targets were located in open ocean, and second, two locations in which targets were embedded within sea ice, and ice floes. All data are quad-polarimetric, with an ascending mode and single-look complex (SLC) format, with three images having an average incidence angle of 39° (in the centre of each image) and one with 31°. The resolution in ground range is 5.1 m, whereas the azimuth resolution is 4.3 m. All data are processed as sigma nought. In our previous paper [22], we focused on the polarimetric behaviour of radar scattering from icebergs. In this work, we focus on target detection as a continuation of our previous work. Table 1 shows the dates of acquisition for each SAR image, and Figure 1 shows the location of acquisition.

**Table 1.** ALOS-2/PALSAR-2 image properties. Latitude and longitude use centre degrees, minutes, seconds (DMS) coordinates. Incidence angle range is min, centre, and max to indicate the near, mid and far range of each image. Ground resolution is ALOS-2 quad-polarimetric mode. Time is UTC. Note that Savissivik was acquired in winter months.

| ID | Location | Lat/Lon (DMS) | Resolution (m) | Incidence Angle Range (°) | Date/Time |
|---|---|---|---|---|---|
| (1) ALOS2066231 360-150815 | Blosseville Coast | 68°02′13.2″ N 30°19′58.8″ W | 4.3 × 5.1 | 37, 39, 41.5 | 15 August 2015 01:26 |
| (2) ALOS2064761 430-150805 | Nuugaatsiaq | 71°25′26.4″ N 53°26′52.8″ W | 4.3 × 5.1 | 37, 39, 41.5 | 5 August 2015 02:48 |
| (3) ALOS2064461 300-150803 | Isortoq | 65°07′08.4″ N 39°13′37.2″ W | 4.3 × 5.1 | 37, 39, 41.5 | 3 August 2015 02:07 |
| (4) ALOS2191031 530-171206 | Savissivik | 75°52′19.2″ N 62°10′48″ W | 4.3 × 5.1 | 29, 31, 33.6 | 6 December 2017 02:52 |

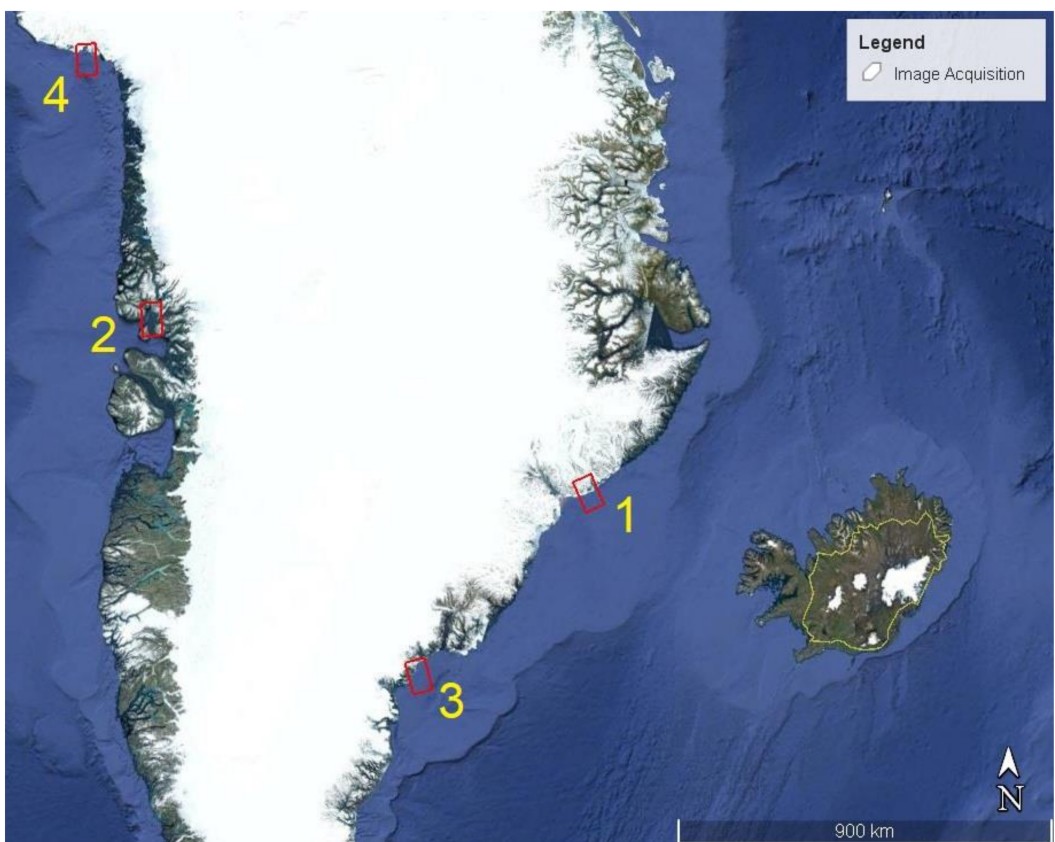

**Figure 1.** Google Earth image of data acquisition. Red boxes indicate the image footprints. Yellow numbers correspond to the image IDs in Table 1. Compass point indicates north. Scale: 900 m. Reprinted with permission from ref. [22] Copyright 2020 Remote Sensing

## 4. Methods

### 4.1. Preliminary Image Analysis

In this section, we present the Pauli RGB images as a reference for the detector analysis. These images were also presented in our previous paper [22]. Figure 2 presents the Pauli RGB for Blosseville, Figure 3 presents Nuugaatsiaq, Figure 4 presents Isortoq and Figure 5 presents Savissivik. RGB images are composed of the three intensities (colours) of the Pauli components: red = *HH + VV*, green = *HH − VV* and blue = *2HV*. Iceberg identification was similar to our previous work [22]. Here, we used visual analysis to identify icebergs based on whether an iceberg was a bright spot on an image, or had a shadow. We also excluded other targets such as ships, islands and buoys from the analysis whether it was suspected that a target was not an iceberg. For example, ships tend to present as elongated targets on an image. The alpha and entropy are calculated using the Cloude–Pottier eigenvalue/eigenvector polarimetric decomposition [41]. In this case, the entropy is calculated as the logarithmic sum of the eigenvalues of the covariance matrix. Entropy refers to the randomness of the scattering behaviour and can be used to estimate whether polarimetric targets are depolarised or have a more dominant scattering mechanism. Alpha is calculated from the eigenvectors of the covariance matrix. It is related to the incidence angle and dielectric constant of the scatterers [42]. Therefore, it can be used to determine a particular scattering mechanism (odd bounce, even bounce, dihedral scattering). Within sea ice, we focused on bright pixels to identify any icebergs embedded within floes. We also indicate the coastline with a red line in the images. It should be noted that in Figure 5, Blosseville Coast S is not included in our analysis. We will call the image for Blosseville

Coast N [22] as Blosseville for short. In Figure 2, a yellow box indicates the area used for detection imaging in Figures 8–10.

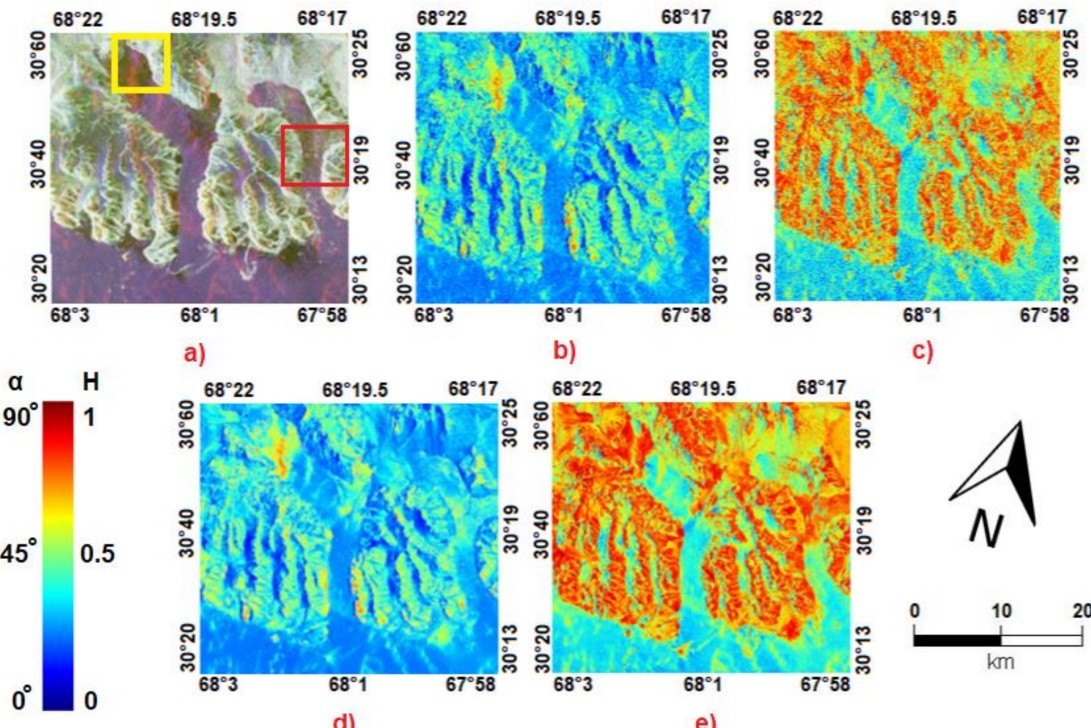

**Figure 2.** Output images for Blosseville, 15 August 2015 01:26: (**a**) Pauli RGB image with extent shown, (**b**) alpha image in a 5 × 5 window, (**c**) entropy image in a 5 × 5 window, (**d**) alpha image in an 11 × 11 window, and (**e**) entropy image in an 11 × 11 window. The red box indicates the extent of previous analysis in this location. The yellow box indicates the area used in the detection tests. Adapted with permission from ref. [22]. Copyright 2020 Remote Sensing.

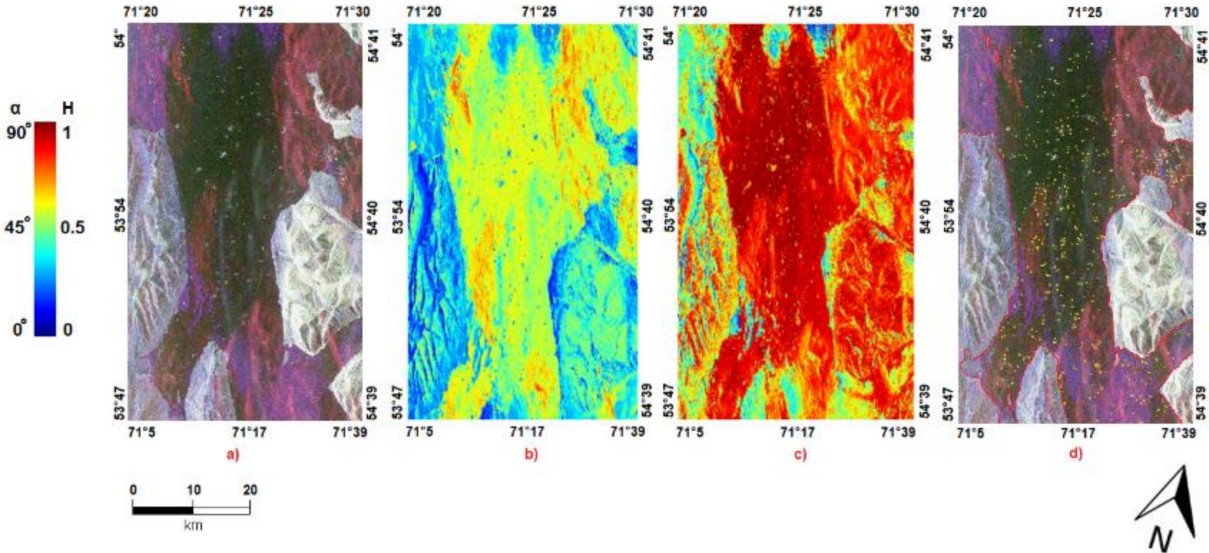

**Figure 3.** Output images for Nuugaatsiaq, 15 August 2015 01:26: (**a**) Pauli RGB image, (**b**) alpha image in a 5 × 5 window, (**c**) entropy image in a 5 × 5 window, and (**d**) visual identification of icebergs and land mask. Yellow dots are icebergs. The red line represents the shoreline. Reprinted with permission from ref. [22] Copyright 2020 Remote Sensing.

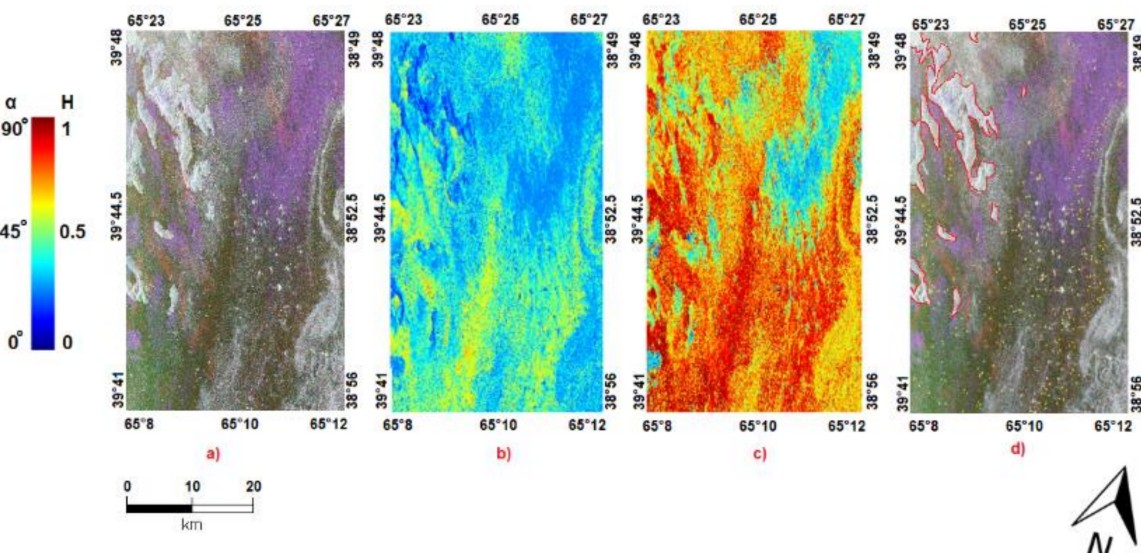

**Figure 4.** Output images for Isortoq, 15 August 2015 01:26, (**a**) Pauli RGB image, (**b**) alpha image in a 5 × 5 window, (**c**) entropy image in a 5 × 5 window, (**d**) visual identification of icebergs and land mask Yellow dots are icebergs. The red line represents the shoreline. Reprinted with permission from ref. [22] Copyright 2020 Remote Sensing.

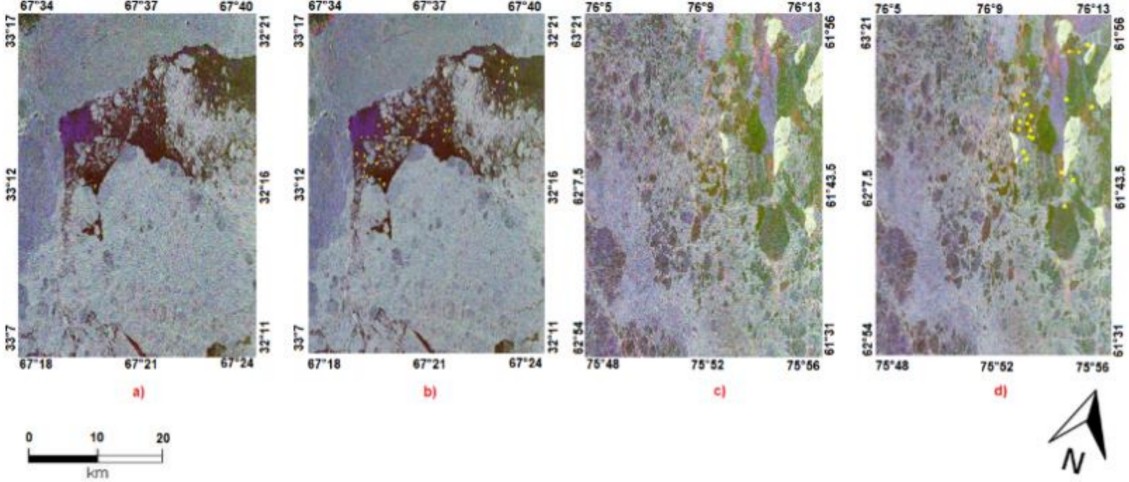

**Figure 5.** Output images for (**a**) Blosseville Coast S Pauli RGB image 20 June 2015 01:26, (**b**) Blosseville Coast S visual analysis, (**c**) Savissivik RGB image 6 December 2017 02:52, and (**d**) Savissivik visual analysis. Yellow dots are icebergs. Note that Blosseville Coast S is not used in this paper. Reprinted with permission from ref. [22] Copyright 2020 Remote Sensing.

### 4.2. Identifying Targets and Clutter

Icebergs were visually identified in images using RGB images in which the contrast could be adjusted accordingly. It is important to note that due to missing in situ validation data, this analysis is restricted to icebergs that can be identified through visual analysis, so we cannot reliably detect smaller icebergs (<120 m). In order to evaluate the statistics for probability of false alarms ($P_F$), we considered areas inside images that were presenting either open ocean or sea ice. Figure 6 shows the clutter area as a red polygon and icebergs as yellow dots for Blosseville. Nuugaatsiaq and Isortoq, and Savissivik. As we can see, the clutter in Blosseville and Nuugaatsiaq is open ocean, while in Isortoq and Savissivik it is sea ice. Note that in Figure 6a, azimuth ambiguities are visible in the centre to the left of the glacier tongues and just below the clutter area. These are caused by the elevation of the mountains by the coastline. It should also be noted that in Isortoq, the icebergs are indicated by blue dots, and in Savissivik, the green areas. In total, we selected 3242 icebergs for this analysis.

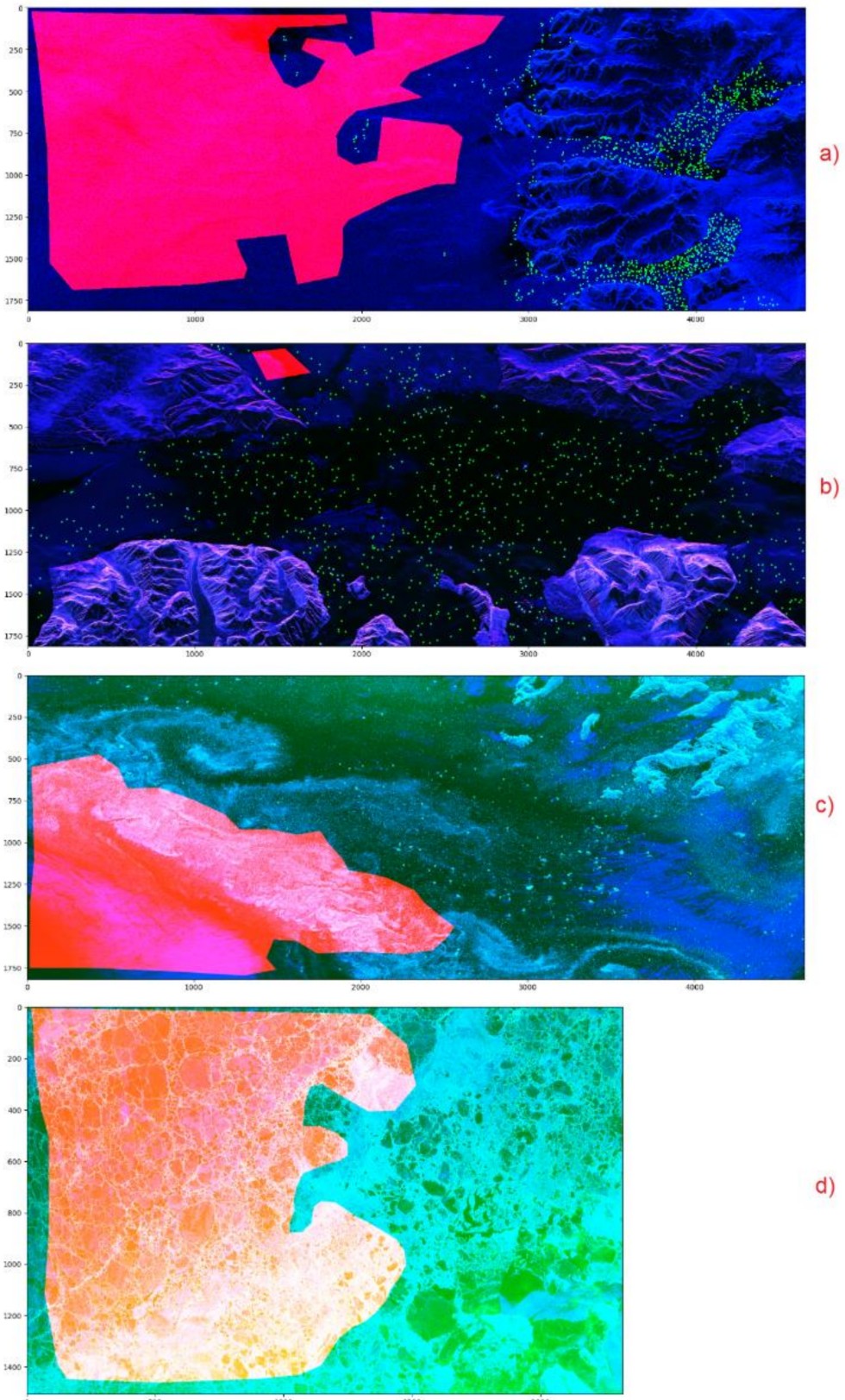

**Figure 6.** Pauli images of each location. (**a**) Target and clutter mask for Blosseville. Scenario is a mix of open ocean, and glacier tongues. Azimuth ambiguities are present in the middle of the image. (**b**) Nuugaatsiaq. Scenario is a mix of open ocean, and islands. (**c**) Isortoq. Scenario is a mix of sea ice and ice floes. Note blue dots for icebergs. (**d**) Savissivik. Scenario is mainly sea ice, with embedded targets, and green areas indicate icebergs. Red polygons indicate clutter.

### 4.3. Preliminary Detection Comparison

Here, we show the comparison of the different detectors in three polarimetric modes, quad-polarisation, dual-polarisation and single-polarisation. We will first present the detector images for a qualitative assessment over a zoomed area and after that, receiver operating characteristic (ROC) curves for a quantitative assessment. Dual-polarimetric detection is achieved by using the channels *HH* and *HV* to work with the detectors designed only for dual-polarimetric data. Single-polarimetric intensities are achieved by focusing on the C11, C22, C33, T11 and T22 elements of the covariance and coherency matrices. All detection images have been created using a window size for the target area or $5 \times 5$ and (when required) a clutter area of $105 \times 105$ pixels using a guard window of $35 \times 35$. Figure 7 represents the guard window approach we used. In the following, we will also test changes in this window configuration.

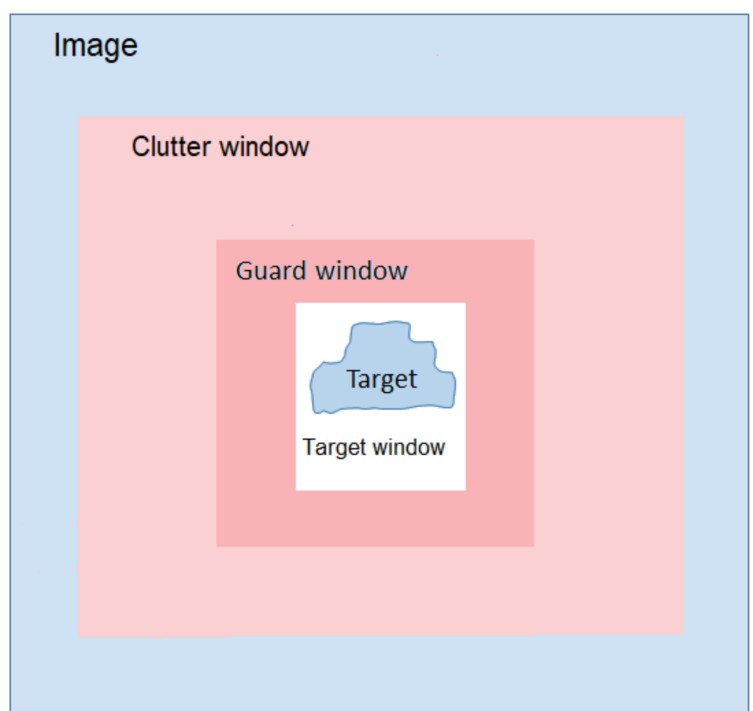

**Figure 7.** The guard window approach. The blue area represents image pixels, the light pink area represents the clutter window, and the dark pink area represents the guard window, to eliminate background clutter from the target window (white). In the case of this analysis, the target is an iceberg.

Detection Distances

Here, we show the images of detection thresholds that are set as distances generated by the detector algorithms. This method allows for a preliminary understanding of the detection capabilities and the contrast without having to select threshold values. We apply threshold values afterwards to generate results in Section 5. Figure 8 shows iDPolRAD, DPolRAD, n filter and symmetry detection images for Blosseville. Please note: land (bottom and top of the image) will be detected too, but we can exclude it using land masks. If we look at the images, we seem to have the clearest detection using the reflection symmetry detector followed by the notch filter and the DPolRAD close behind. The next images in Figure 9 show parameters from the Cloude–Pottier decomposition, with alpha, entropy, first eigenvalue ($\lambda_1$) and third eigenvalue ($\lambda_3$) [6].

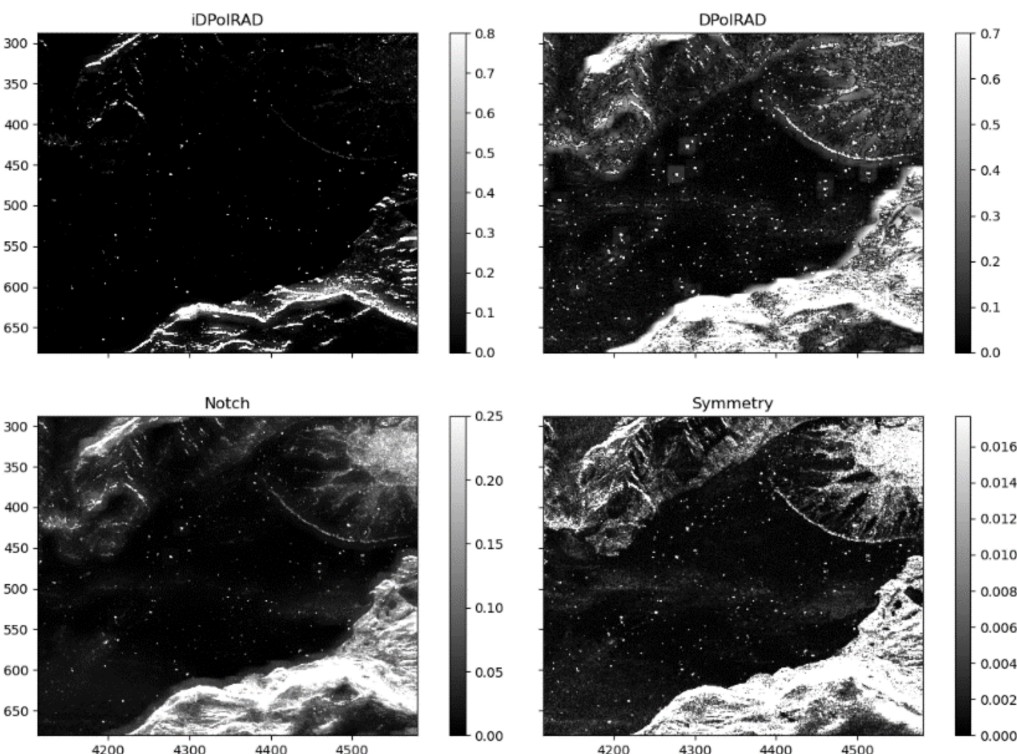

**Figure 8.** Iceberg detection iDPolRAD, DPolRAD, notch filter and symmetry for Blosseville. Image size is $350 \times 500$ pixels. Greyscales indicate detection intensity.

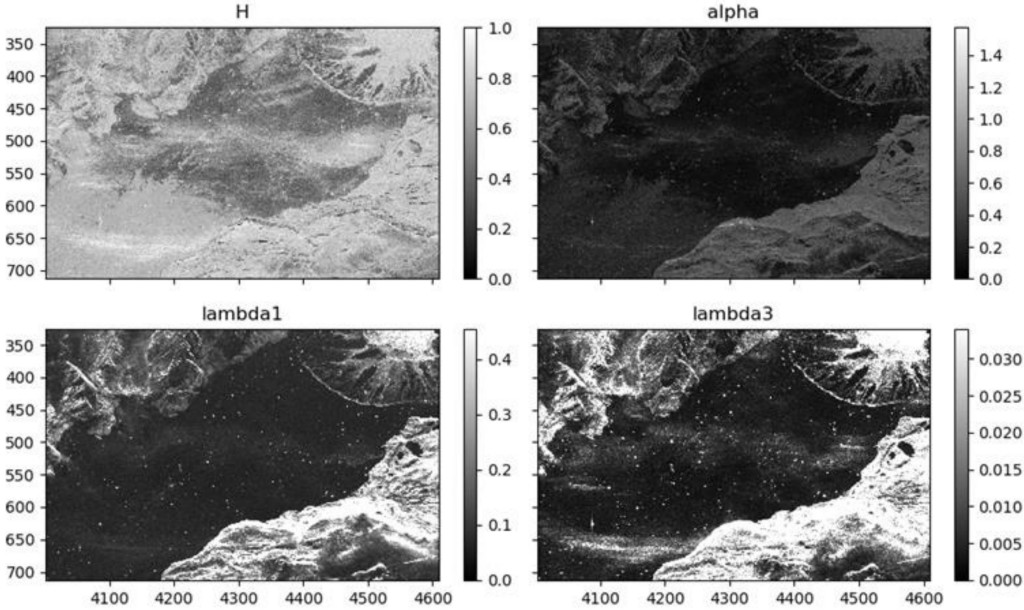

**Figure 9.** Iceberg detection entropy H, alpha, $\lambda_1$ and $\lambda_3$ for Blosseville. Image size is $350 \times 600$ pixels. Greyscales indicate detection intensity.

From the images, it is clear that the alpha and entropy are not perfect detectors. Rather, they are better to show classification of sea ice as in Section 4.1. $\lambda_1$ and $\lambda_3$ seem to be more suited for the detection task. The images in Figure 10 show the first ($\sigma_1$) and third ($\sigma_3$) eigenvalues from the PMF, then the PWF and the OPD.

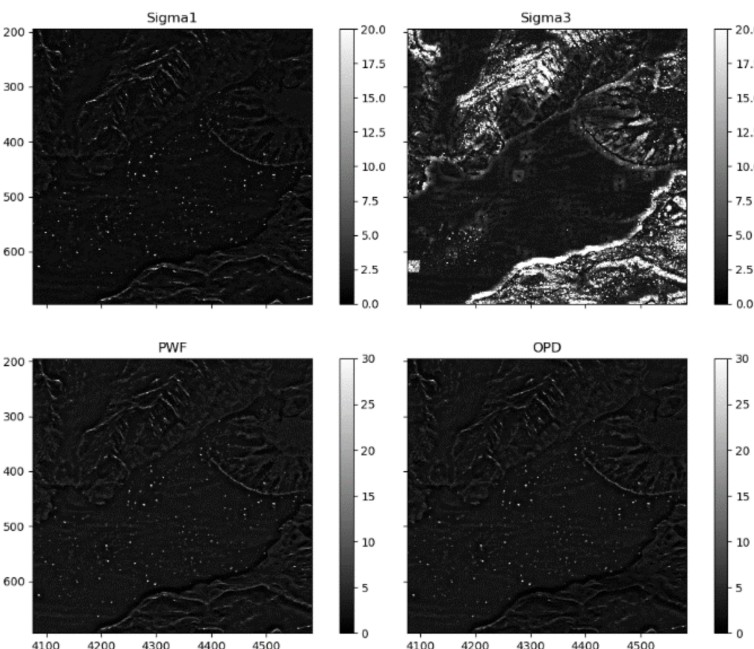

**Figure 10.** Iceberg detection $\sigma_1$, $\sigma_3$, the PWF, and the OPD for Blosseville. Image size is $400 \times 600$ pixels. Colourmaps indicate detection intensity.

From the images, it is clear that $\sigma_1$, the PWF and the OPD detectors tend to give the visually best performance, while $\sigma_3$ is missing many icebergs. Detector output shows similar trends in other locations. It is clear that a quantitative analysis is needed to evaluate the performance in a better way. This follows in the next section.

## 5. Results

In this section, we present the ROC curves. These are plots of probability of detection ($P_D$) against probability of false alarms ($P_F$). We first present the quad-polarimetric results, and then move to dual-polarimetric and single-polarimetric results. We will first show the ROC curves and then summarise them in tables in next section.

### 5.1. Quad-Polarimetric ROC Curves

Figure 11 presents a ROC curve for the case of open ocean clutter (Blosseville and Nuugaatsiaq) using a target window size of $5 \times 5$ and a clutter window size of $105 \times 105$.

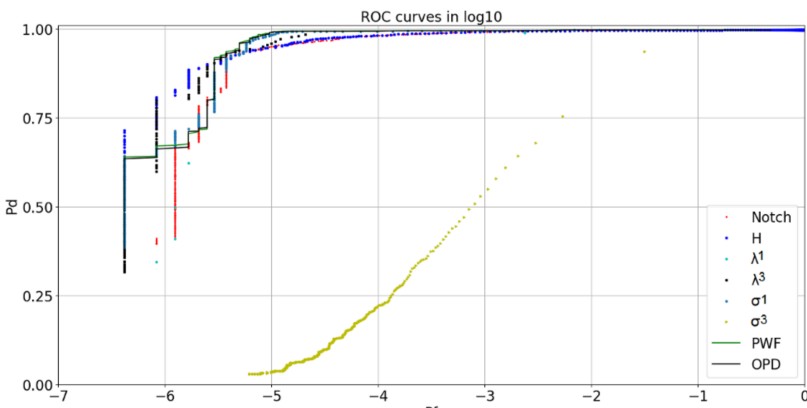

**Figure 11.** Iceberg detection ROC curves for open ocean, target size $5 \times 5$, clutter size $105 \times 105$. Notch means the polarimetric notch filter, H means entropy, $\lambda_1$ and $\lambda_3$ are the first and third eigenvalues of the covariance matrix, $\sigma_1$ and $\sigma_3$ are from the polarimetric match filter, PWF means the polarimetric whitening filter, and OPD means the optimal polarimetric detector.

From Figure 11, we can see that in case of open ocean, $\sigma_3$ of the PMF does not perform properly. This is expected, since icebergs are brighter than the background in most open ocean conditions. All the other detectors provide a similar performance, but we can see some differences when we consider different levels of false alarm rates.

(1) $P_F = 10^{-5}$

Here, we can see that the PWF and the OPD both seem to provide the best performance (0.990) followed by $\sigma_1$, the notch filter, entropy, $\lambda_3$ and $\lambda_1$. The worst performance is $\sigma_3$ of the PMF, with a $P_D$ of 0.034.

(2) $P_F = 10^{-6}$

Here, the best detector is the entropy (0.808) followed by $\lambda_3$, the PWF, $\sigma_3$, the OPD, and the notch filter. The worst performance is $\lambda_1$ (0.345).

Figure 12 presents a ROC for the case of sea ice (Isortoq and Savissivik).

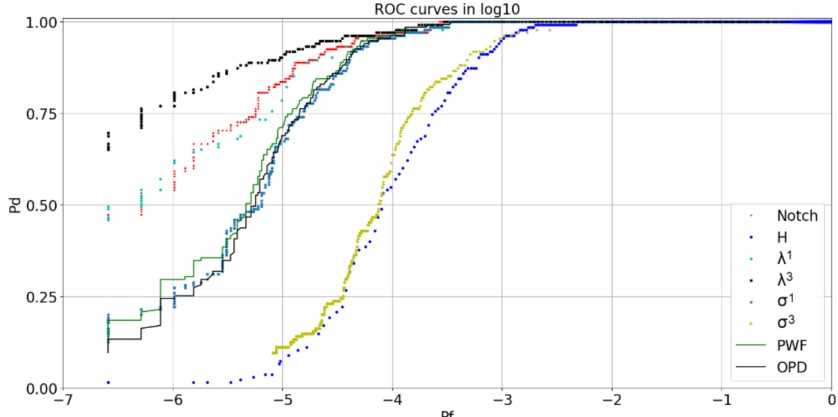

**Figure 12.** Iceberg detection ROC curves for sea ice, target window $5 \times 5$, clutter window $105 \times 105$. Notch means the polarimetric notch filter, H means entropy, $\lambda_1$ and $\lambda_3$ are the first and third eigenvalues of the covariance matrix, $\sigma_1$ and $\sigma_3$ are from the polarimetric match filter, PWF means the polarimetric whitening filter, and OPD means the optimal polarimetric detector.

Interestingly, we can see that in Figure 15, the entropy performance declines, which is most likely due to the number of scatterers present in the sea ice, which increases their entropy. Other adaptive detectors (e.g., the PMF, the PWF, and the PNF) also show a decreased performance in sea ice, since they are detecting heterogeneity of the sea ice.

(1) $P_F = 10^{-5}$

The best detector is the $\lambda_3$ eigenvalue with $P_D$ equal to 0.770. This is followed by the notch filter, $\lambda_1$, the PWF, the OPD, $\sigma_1$ and $\sigma_3$. The worst performance is the entropy (0.074).

(2) $P_F = 10^{-6}$

Here, the performance degrades, but $\lambda_3$ remains the best detector, with a $P_D$ of 0.770. This is followed by $\lambda_1$, the notch filter, the PWF, the OPD, and $\sigma_1$, with no result for $\sigma_3$. The entropy shows the worst overall performance, with a $P_F$ of 0.015.

We now want to test in a different clutter window. We show the quad-polarimetric ROC curves with a target window of $15 \times 15$ and a train window of $255 \times 255$ window, including a guard of $205 \times 205$ pixels (Figures 13 and 14).

It is interesting to see that by using a larger window, the performance has been reduced. This is because the icebergs analysed here are particularly small (<120 m) and a larger window seems to impede detection. For instance, in open oceans, some of the smaller icebergs can only be detected with probability of false alarms proximal to 1.

Finally, we want to show the effects of NOT using a guard window for the detectors that need training. These detectors may be simpler to implement. This is shown in Figures 15 and 16. The performances are again clearly reducing, showing the importance of using a guard window for training.

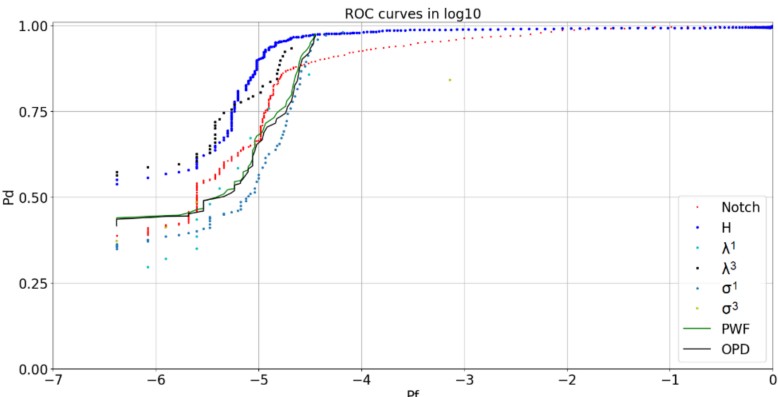

**Figure 13.** Iceberg detection ROC curves for open ocean, target window $15 \times 15$, clutter window $255 \times 255$ with a guard window of $205 \times 205$. Notch means the polarimetric notch filter, H means entropy, $\lambda_1$ and $\lambda_3$ are the first and third eigenvalues of the covariance matrix, $\sigma_1$ and $\sigma_3$ are from the polarimetric match filter, PWF means the polarimetric whitening filter, and OPD means the optimal polarimetric detector.

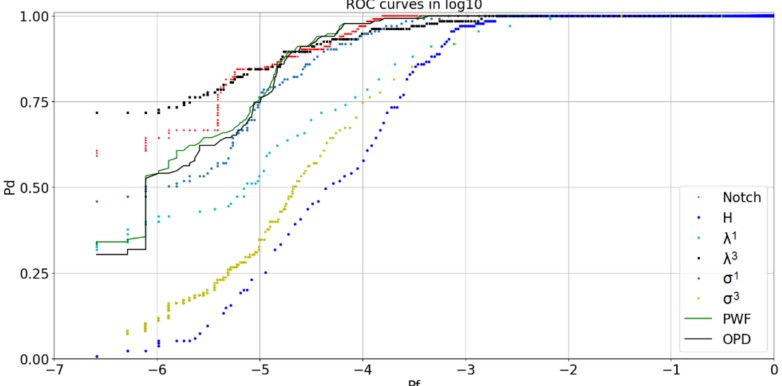

**Figure 14.** Iceberg detection ROC curves for sea ice, target window $15 \times 15$, clutter window $255 \times 255$ with a guard window of $205 \times 205$. Notch means the polarimetric notch filter, H means entropy, $\lambda_1$ and $\lambda_3$ are the first and third eigenvalues of the covariance matrix, $\sigma_1$ and $\sigma_3$ are from the polarimetric match filter, PWF means the polarimetric whitening filter, and OPD means the optimal polarimetric detector.

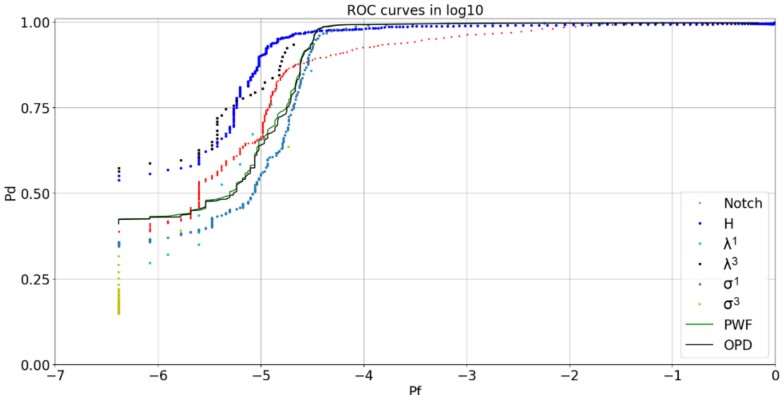

**Figure 15.** Iceberg detection ROC curves for open ocean, target window $5 \times 5$, clutter window $105 \times 105$ with no guard window. Notch means the polarimetric notch filter, H means entropy, $\lambda_1$ and $\lambda_3$ are the first and third eigenvalues of the covariance matrix, $\sigma_1$ and $\sigma_3$ are from the polarimetric match filter, PWF means the polarimetric whitening filter, and OPD means the optimal polarimetric detector.

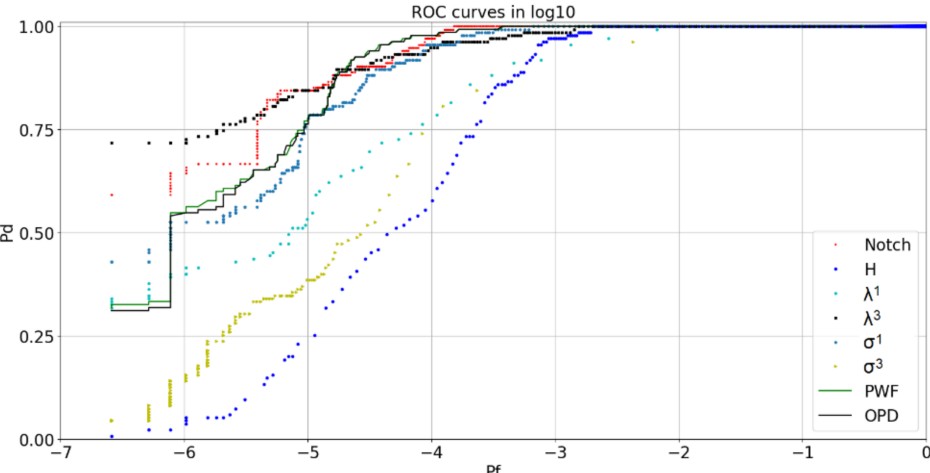

**Figure 16.** Iceberg detection ROC curves for sea ice, target window $5 \times 5$, clutter window $105 \times 105$ with no guard window. Notch means the polarimetric notch filter, H means entropy $\lambda_1$ and $\lambda_3$ are the first and third eigenvalues of the covariance matrix, $\sigma_1$ and $\sigma_3$ are from the polarimetric match filter, PWF means the polarimetric whitening filter, and OPD means the optimal polarimetric detector.

*5.2. Dual-Polarimetric ROC Curves*

For the dual-polarimetric ROC curves, we present the same window size with a guard window, but this time we are only considering three elements of the covariance matrix C11, C22 and C12 for analysis. This means that only the images *HH* and *HV* are used. Figure 17 shows a dual-polarimetric ROC curve with a $5 \times 5$ target window, a $35 \times 35$ guard window size and a $105 \times 105$ clutter window size. All the dual-polarimetric detectors tested provide relatively similar performances. Again, we want to analyse performances using two different levels of false alarm rates.

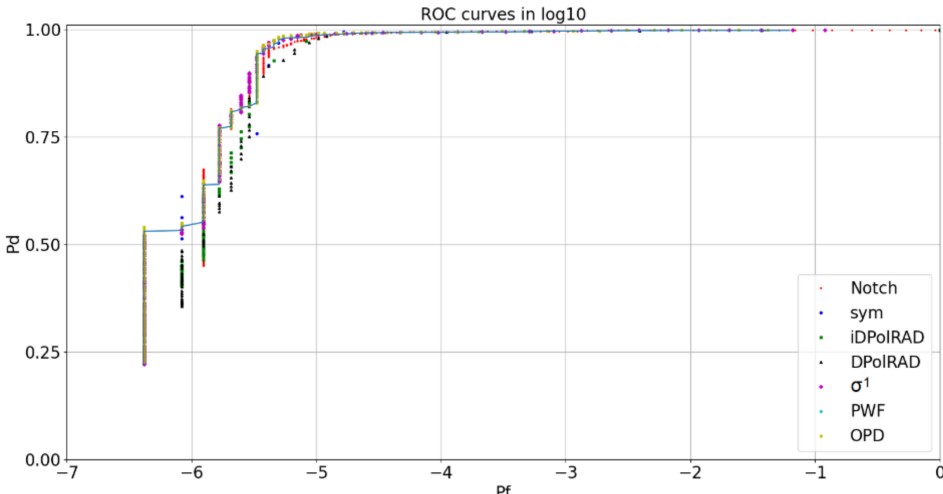

**Figure 17.** Iceberg detection dual-polarimetric ROC curves for open ocean, $5 \times 5$, clutter window size $105 \times 105$. Sym means symmetry detector. iDPolRAD and DPolRAD mean intensity dual-polarisation ratio anomaly detector, and $\sigma_1$ is from the polarimetric match filter.

(1) $P_F = 10^{-5}$

The PWF provides the best performance (0.990) followed by $\sigma_1$, the OPD, the notch and both the DPolRAD and the iDPolRAD. The reflection symmetry detector has the worst performance (0.969). No performance is shown from $\sigma_3$ because in dual-polarimetric detection, we do not have the third eigenvalues.

(2) $P_F = 10^{-6}$

When compared to the quad-polarimetric results, we can see a substantial difference in $P_D$ values. The reflection symmetry detector becomes the best detector (0.612) followed by the PWF, the OPD, $\sigma_1$, the DPolRAD, the iDPolRAD and the notch filter with the worst performance (0.451).

Figure 18 presents the same result with sea ice clutter. Note that we can see differences between quad and dual for all levels of $P_F$.

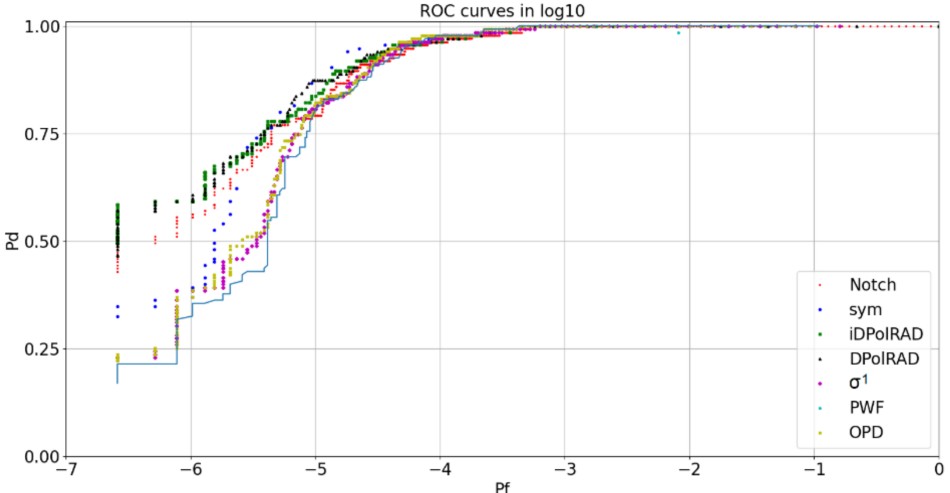

**Figure 18.** Iceberg detection dual-polarimetric ROC curves for sea ice, $5 \times 5$, clutter window size $105 \times 105$. Sym means symmetry detector. iDPolRAD and DPolRAD mean intensity dual-polarisation ratio anomaly detector, $\sigma_1$ is from the polarimetric match filter.

(1) $P_F = 10^{-5}$

Here, we can see that the DPolRAD provides the best performance (0.874) followed by the symmetry, the iDPolRAD, the PWF, $\sigma_1$, and the OPD. The notch filter has the worst performance (0.793).

(2) $P_F = 10^{-6}$

Here, we can see that there is a range of $P_D$ values for all the detectors. The DPolRAD and the iDPolRAD show the best performance (0.593). This is followed by the notch filter and $\sigma_1$, PWF and symmetry. The OPD detector appears to show the worst performance, with a $P_D$ value of 0.313.

The first thing to notice from Figures 17 and 18 is that the notch filter gives an improved performance in sea ice, to the quad-polarimetric version, which helps understand how the improved power of some adaptive detectors may produce several false alarms when there is sea ice. Performance between other detectors is also similar, with, PWF showing the best performance (0.990) in open ocean when $P_F$ is $10^{-5}$. The DPolRAD slightly outperforms with a $P_D$ of 0.874 in sea ice when $P_F$ is $10^{-5}$. As an important finding, we show that the dual-polarimetric performance is overall lower than the quad-polarimetric performance.

### 5.3. Intensities ROC Curves

In this section, we investigate detectors set with intensity of backscattering representing an appropriate scattering mechanism. These certainly include the single-polarimetric detectors when we use the co-polarisation or cross-polarisation channels on their own. However, they also include quad-polarimetric detectors when we consider coherence combinations of co-polarisation channels. We show ROC curves in Figures 19 and 20 using C11, C22 and C33, plus T11 and T22 images. We apply to a window of $5 \times 5$. Again, we want to analyse performances using two different levels of false alarm rates.

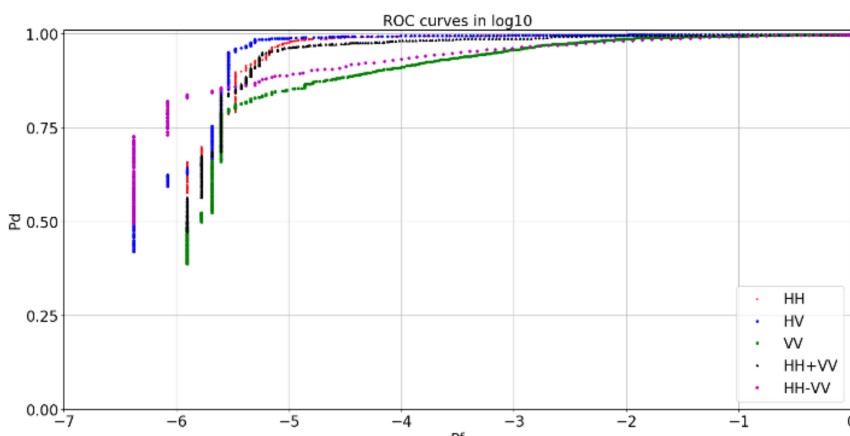

**Figure 19.** Iceberg detection intensity ROC curves for open ocean, $5 \times 5$, clutter window size $105 \times 105$. C11 is a HH polarisation, C22 is a cross-polarised HV polarisation, C33 is a VV polarisation, T11 is a HH + VV polarisation and T22 is a HH − VV polarisation.

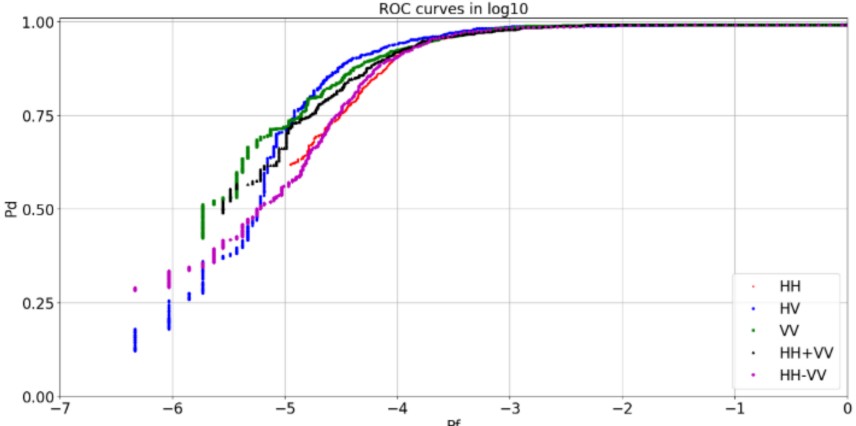

**Figure 20.** Iceberg detection intensity ROC curves for sea ice, $5 \times 5$, clutter window $105 \times 105$. C11 is a HH polarisation, C22 is a cross-polarised HV polarisation, C33 is a VV polarisation, T11 is a HH + VV polarisation and T22 is a HH − VV polarisation.

(1) $P_F = 10^{-5}$

The *HV* cross-channel seems to provide the best performance (0.989) followed by *HH*, *HH + VV* and *HH − VV*. The *VV* channel gives the worst performance (0.851). In open ocean, there is not a significant difference in performance compared to the dual-polarimetric results.

(2) $P_F = 10^{-6}$

Here, the performance degrades and *HH − VV* shows the best performance (0.822) followed by *HH*, and *HV* with the worst performance (0.624). Now the difference is much more apparent in all the channels except *HH − VV*, but this requires quad-polarimetric analysis as above.

Figure 20 shows the ROC performance in sea ice clutter. Again, we want to analyse performances using two different levels of false alarm rates.

(1) $P_F = 10^{-5}$

Here, we can see that the *VV* channel provides the best performance (0.719) followed in order by *HV*, *HH + VV* and *HH − VV* with the worst performance (0.561). No matter what $P_F$ value, the performance has declined.

(2) $P_F = 10^{-6}$

The performance here has degraded, with *HH − VV* providing the best performance (0.334) followed only by HV (0.255).

### 5.4. Best Detectors

Tables 2 and 3 summarise the results and reveal that the best detectors for quad-polarimetric appear to be the PWF, the OPD and $\sigma^1$ detectors, with the worst performance from the $\sigma^3$ in open ocean, and entropy in sea ice. Tables 4 and 5 show that the best detectors for dual-polarimetric detection again appear to be the OPD and the PWF in open ocean. However, for sea ice, the best detectors appear to be the reflection symmetry detector and the DPolRAD. Finally, Tables 6 and 7 show that the best detection intensity appears to be the $HV$ channel when the detection is easier and the $HH - VV$ channel in more complex environments. Note that numbers in green represent the highest $P_D$ values after fixing the $P_F$.

**Table 2.** Probabilities of detection for quad-polarimetric open ocean. Bold numbers indicate highest $P_D$ values.

| | $\lambda_1$ | $\lambda_3$ | H | PNF | $\sigma_1$ | $\sigma_3$ | PWF | OPD |
|---|---|---|---|---|---|---|---|---|
| $P_F = 1^6$ | 0.345 | 0.800 | **0.809** | 0.412 | 0.668 | | 0.671 | 0.663 |
| $P_F = 1^5$ | 0.799 | 0.944 | 0.948 | 0.950 | 0.983 | 0.034 | **0.990** | **0.990** |
| $P_F = 1^4$ | N/A | 0.986 | 0.982 | 0.986 | 0.994 | 0.218 | **0.995** | 0.994 |

**Table 3.** Probabilities of detection for quad-polarimetric sea ice. Bold numbers indicate highest $P_D$ values.

| | $\lambda_1$ | $\lambda_3$ | H | PNF | $\sigma_1$ | $\sigma_3$ | PWF | OPD |
|---|---|---|---|---|---|---|---|---|
| $P_F = 1^6$ | 0.570 | **0.770** | 0.015 | 0.541 | 0.222 | | 0.296 | 0.244 |
| $P_F = 1^5$ | 0.785 | **0.904** | 0.074 | 0.837 | 0.674 | 0.111 | 0.711 | 0.689 |
| $P_F = 1^4$ | **0.970** | **0.970** | 0.548 | **0.970** | 0.956 | 0.615 | 0.963 | 0.963 |

**Table 4.** Probabilities of detection for dual-polarimetric open ocean. Bold numbers indicate highest $P_D$ values.

| | iD | D | PNF | $\sigma_1$ | Sym | PWF | OPD |
|---|---|---|---|---|---|---|---|
| $P_F = 1^{-6}$ | 0.462 | 0.487 | 0.451 | 0.535 | **0.612** | 0.551 | 0.542 |
| $P_F = 1^{-5}$ | 0.972 | 0.970 | 0.978 | 0.988 | 0.969 | **0.990** | 0.985 |
| $P_F = 1^{-4}$ | | 0.993 | **0.994** | 0.993 | **0.994** | **0.994** | 0.993 |

**Table 5.** Probabilities of detection for dual-polarimetric sea ice. Bold numbers indicate highest $P_D$ values.

| | iD | D | PNF | $\sigma_1$ | Sym | PWF | OPD |
|---|---|---|---|---|---|---|---|
| $P_F = 1^{-6}$ | **0.593** | **0.593** | 0.556 | 0.385 | 0.363 | 0.370 | 0.313 |
| $P_F = 1^{-5}$ | 0.837 | **0.874** | 0.793 | 0.808 | 0.867 | 0.815 | 0.800 |
| $P_F = 1^{-4}$ | 0.963 | 0.963 | 0.963 | 0.970 | 0.970 | **0.978** | **0.978** |

**Table 6.** Probabilities of detection for intensities open ocean. Bold numbers indicate highest $P_D$ values.

| | HV | HH | VV | HH + VV | HH-VV |
|---|---|---|---|---|---|
| $P_F = 1^{-6}$ | 0.624 | 0.782 | | | **0.822** |
| $P_F = 1^{-5}$ | **0.989** | 0.977 | 0.851 | 0.962 | 0.890 |
| $P_F = 1^{-4}$ | **0.995** | 0.993 | 0.911 | 0.982 | 0.930 |

**Table 7.** Probabilities of detection for intensities sea ice. Bold numbers indicate highest $P_D$ values.

|  | HV | HH | VV | HH + VV | HH − VV |
|---|---|---|---|---|---|
| $P_F = 1^{-6}$ | 0.255 |  |  |  | **0.334** |
| $P^F = 1^{-5}$ | 0.706 |  | **0.719** | 0.663 | 0.561 |
| $P_F = 1^{-4}$ | **0.940** | 0.910 | 0.926 | 0.920 | 0.905 |

## 6. Discussion

In this section, we outline and evaluate the results of the analysis.

Using a target window of $5 \times 5$ and clutter window of $105 \times 105$, the ROC curves suggest that the best performance for detecting icebergs in open ocean provides a $P_D$ of 0.99 with a $P_F$ of $10^{-5}$ or 0.81 for $P_F$ $10^{-6}$. When we compare with iceberg detection in sea ice with the same false alarm rate, we have a $P_D$ of 0.9 and 0.77, respectively, indicating that detection in sea ice is more complex. This can be explained by the fact that sea ice can trigger false alarms, especially in more powerful detectors [13].

Interestingly, we can see that performance evaluation in dual-polarimetric detectors shows a reduction in $P_D$ for small values of false alarms. For instance, fixing $P_F$ to $10^{-6}$, the $P_D$ is 0.61 in open ocean and 0.59 in sea ice. This is in line with the fact that by reducing the number of polarisation channels, our discrimination capability is reduced. However, in open ocean, the detection task is easier and we use larger values of $P_F$. As a result, the difference in performance between dual and quad detection is very limited.

Results from detectors based on intensities show again that in easier scenarios (such as in open ocean), where $P_F = 10^{-5}$, the difference is minor, with *HV* reaching a $P_D$ of 0.985 (compared to 0.99 for quad). However, when more complex scenarios are considered (such as in sea ice), the performance of *HV* degrades and a quad-polarimetric detector performs better. For instance, with $P_F = 10^{-6}$ in open ocean, the *VV* channel shows $P_D = 0.78$ (compared to 0.99 of quad); and within sea ice in the same channel, the $P_D$ values are 0.71 for $P_F = 10^{-5}$ (compared to 0.90 of quad) and 0.25 for $P_F = 10^{-6}$ (compared to 0.77 of quad).

To summarise, when the detection task is easier because larger icebergs in open ocean are easier to visualise, a single-polarisation channel will perform relatively similarly to a quad-polarimetric one. However, for the detection of icebergs in sea ice or for smaller icebergs (smaller false alarm rate), adding polarimetric information will help. Finally, we saw that when we increase the target window to $15 \times 15$ and clutter window to $255 \times 255$, the $P_D$ value reduces. This is in line with the fact that when we use a bigger window, more pixels are included, thus the target is smeared (when averaging), and the probability of detection is reduced. Moreover, when ring windows are not used, then the results show a degradation of performance. The absolute values for $P_D$ need to be taken with care since we only restricted the analysis to the icebergs that are clearly identifiable. However, the relative comparison of detectors should still be preserved if fainter icebergs were introduced. It should also be noted that additional data to determine whether detection is impeded by temperature or precipitation in winter and summer was not available.

## 7. Conclusions

In this work, we tested an ALOS-2 dataset with six state-of-the art detectors, most of them designed to be used for ship detection. The detectors are the intensity dual-polarisation ratio anomaly detector (iDPolRAD), the polarimetric notch filter (PNF), the polarimetric match filter (PMF), the reflection symmetry detector, the multi-look polarimetric whitening filter (MPWF) and the optimal polarimetric detector (OPD). We estimated detection performance over four ALOS-2 quad-polarimetric single-look complex SAR images, in four locations in Greenland. To compare the performances of each detector, we performed the analysis using two scenarios. Two of the images had icebergs in an open ocean setting, and two showed icebergs embedded within sea ice. In total, we considered 3,242 icebergs in this analysis. We show that, overall, the quad-polarimetric detectors, the OPD and the PWF, provide the best detection performance, with a $P_D$ of 0.99 in open ocean

when the false alarm is set to $10^{-5}$. However, in sea ice, $\lambda_3$ shows the best performance (0.90). For dual-polarimetric performance, we conclude that the PWF gives the best performance in open ocean (0.90). In the sea ice, the best detector is the DPolRAD (0.978). Finally, we tested the data in the single-polarisation mode, showing that the best performance is *HV* in the open ocean (0.99) and in the sea ice (0.87). The differences between quad- and dual- or single-polarimetric detectors are more evident with low detection probabilities set at $P_F = 10^{-6}$ when compared to the *HV* channel as quad-polarimetric detectors increase the $P_D$ from 0.62 to 0.81 for open ocean and from 0.26 to 0.77 for sea ice.

**Author Contributions:** Conceptualization, J.B. and A.M.; methodology, J.B.; software, A.M.; validation, J.B., A.M. and V.A.; formal analysis, J.B.; investigation, J.B; resources, A.M.; data curation, J.B.; writing—original draft preparation, J.B.; writing—review and editing, J.B. and V.A.; visualization, J.B.; supervision, A.M.; project administration, J.B. All authors have read and agreed to the published version of the manuscript.

**Funding:** This research received no external funding.

**Acknowledgments:** All data were provided by the project number 1151. ALOS-2 Product-JAXA 2017, all rights reserved.

**Conflicts of Interest:** The authors do not declare any conflicts of interest.

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
