# Peer review of "Comparison of Target Detectors to Identify Icebergs in Quad-Polarimetric L-Band Synthetic Aperture Radar Data"

_remotesensing, doi:10.3390/rs13091753_

Round 1

Reviewer 1 Report

This is potentially a useful piece of research but the presentation is too unclear for it to be meaningfully assessed yet. It seems like a first draft of a manuscript that wasn’t ready to be submitted. I am afraid I stopped reviewing at the end of section 4, i.e. I did not try to review the results section, because I felt that I hadn't understood enough of the methods. The manuscript needs a more systematic introduction, and - crucially - a systematic mathematical formulation so that the reader can meaninully compare the different approaches that are being tested here. There is possibly some unnecessary matematical detail too. The abstract contains too much technical detail. The introduction begins with some very broad statements, not supported by references, then rather abruptly switches to quite technical detail. Some restructuring would be helpful – statements about morphologies of icebergs and scattering mechanisms, should be grouped together and presented in that order. As it is, they appear throughout the text. How much of the mathematical development of polarimetry is actually needed to support the subsequent argument?

The general mathematical formulation of the detectors needs a bit of explanation. To take eq (5) as an example, it is written as Λ = (equation) > T. I think we should interpret this as meaning Λ is defined as (equation), and if Λ > T, an iceberg has been detected.

Detailed comments (only go as far as section 4)

11 ‘small sized’ - say what this means

15 (and many times subsequently) ‘pol’ → ‘polarisation’

16 ‘naught’ → ‘nought’

18 ‘by reducing the channels to HH and VV’ this is not intelligible

18/19 ‘by using intensity channels.. T22’ again – the reader can not understand what this means

38 ‘Smaller icebergs are larger hazards...’ How small? Does this inverse relationship between size and hazard extend to the smallest icebergs? Please give a reference.

44 ‘polarimetric behaviour of targets’ Well, I suppose I know what you mean by this phrase but it’s not really precise.

53 ‘appear brighter’ → ‘have higher backscatter’

54 ‘Previous work found...’ The connection of these two sentences to the previous text is not explained.

56 ‘...and then the sea clutter. - or, I assume, in the opposite order?

63 ‘This is likely due to… toppling over’ I don’t understand why this would be true.

65 ‘pixel sizes are smaller (1-4 pixels)’ Smaller than what? And what does 1-4 pixels mean?

68 ‘higher’ → ‘larger’

73 ‘on ALOS...’ → ‘applied to ALOS...’?

81 ‘with PolSAR for icebergs in L band’ → ‘applied to L-band PolSAR data’ (perhaps – the meaning isn’t clear)

81 ‘PolSAR’ - is this a typo for PALSAR? Or an unexplained abbreviation for ‘polarimetric SAR’?

92 ‘Where’ doesn’t begin a new sentence, so don’t capitalise it. (Happens elsewhere too.)

92 ‘… H is a linear horizontal wave’ This paragraph needs to be worded more carefully. The reader doesn’t know whether HV etc represents an amplitude, intensity, backscatter coefficient, or what.

95 ‘deterministic polarimetric target’ - what is this?

102 ‘three-dimensional’ - but in eq. (2) you show it as four-dimensional.

104 ‘on the ground’ → ‘on the ocean surface’??

105 ‘nondeterministic’ - what does this mean here?

108 ‘partial target’ - what is this?

110 you define the matrix C but not T.

113 ‘averaging’ - over what?

117 what do the lower case ‘s’ refer to in eq. (4)? Are these the same terms as in eq (1) (where you used upper case?)

131 ‘review the literature to which they have been applied’ This isn’t what you mean. Surely ‘review the literature that describes their application’, or something like that?

139 ‘HV and VH (or HH, VV)’ very confusing.

140 ‘Due to volume scattering...’ this sentence belongs in the introduction, not here.

145 ‘sizes… are spatial[ly] averages’ This isn’t what you mean. Please reword it.

145 ‘spatially’ → ‘spatial’

152 ‘if there is a reduction in volume’ - of what?

152 ‘The complex nature of icebergs...’ this belongs in the introduction.

155 ‘The detector increased the contrast...’ How is this defined? How is reduction in clutter defined?

167 Presumably Λ and T here are not the same as in eq (5)?

172 ‘on if’ → ‘on whether’, or just ‘whether’, would be more idiomatic here I think

175-179 I don’t understand the relation between eqq (7) and (8)

182 ‘the area between the testing...’ I think a diagram would help enormoulsy here!

200 Why has the symbol for a detector been changed from lambda to gamma here? Does it signifiy something different?

205 ‘As this algorithm is quite general’ In what way is it more general than the others?

208 ‘For the sake of brevity’ ?? I don’t understand. You present another equation that differs from the previous one only by the substitution of one variable for another.

220 ‘contract’ → ‘contrast’?

225 Please explain eq (12) more.

248 What are X and Σ in eq (14)?

267 What are r and Y? (I now see they are defined later. Please move the definitions nearer to the equations that use them).

268 is ‘tr’ here different from ‘Trace’ in eq (2)?

269 ‘It is suggested… known to obey’ If it is known to follow a gamma distribution, the suggestion is unnecessary. If it has to be suggested, it can’t be known.

288 ‘;’ → ‘,’

294 ‘three images… average incidence angle’ not clear what this means – what are you averaging over?

296 ‘naught’ → ‘nought’

297 ‘polarimetric behaviour of icebergs’ → ‘polarimetric behaviour of radar scattering from icebergs’

303 Table 1. Why does each image have three incidence angles?

317 ‘on if’ → ‘on whether’

319 ‘if it was suspected… not an iceberg’ How can you tell?

322 ‘We present Blosseville Coast.. as Blosseville’ - I don’t understand this statement.

328 alpha, entropy haven’t been defined. How were these calculated?

358 ‘we exclude smaller icebergs (<100 m).’ I don’t understand. If you know that they are icebergs, why exclude them? Maybe you mean that you couldn’t reliably detect icebergs smaller than 100 m?

372 ‘yellow areas’ - Please identify – I can’t see any!

373 ‘Note the transparency of the polygons does not affect the analysis.’ I don’t understand this remark.

378 ‘reducing the channels to HH and HV’ - I don’t understand what you mean.

387 ‘an image with pixels’ - are there other kinds?

388 ‘train window’ - please could you refer to it as a ‘training window’? ‘Train window’ conjures up entirely the wrong mental image!

394 ‘, too’ → ‘too,’

397 ‘Cloude-Pottier decomposition’ - I think it’s the first we’ve heard about this. Please include in methods.

407-420 ‘it is clear...tend to give visually...’ It’s odd that a paper which has given such a lot of mathematical detail earlier on should rely on quite qualitative assessments of performance. This remains an issue through the results section.

Author Response

This is potentially a useful piece of research but the presentation is too unclear for it to be meaningfully assessed yet. It seems like a first draft of a manuscript that wasn’t ready to be submitted. I am afraid I stopped reviewing at the end of section 4, i.e. I did not try to review the results section, because I felt that I hadn't understood enough of the methods. The manuscript needs a more systematic introduction, and - crucially - a systematic mathematical formulation so that the reader can meaninully compare the different approaches that are being tested here. There is possibly some unnecessary matematical detail too. The abstract contains too much technical detail. The introduction begins with some very broad statements, not supported by references, then rather abruptly switches to quite technical detail. Some restructuring would be helpful – statements about morphologies of icebergs and scattering mechanisms, should be grouped together and presented in that order. As it is, they appear throughout the text. How much of the mathematical development of polarimetry is actually needed to support the subsequent argument? – Thank you for the overall comment. We have now updated the introduction and completely updated the abstract to remove the technical detail. We have added references to the beginning of the introduction and moved sections of text to appropriate parts of the manuscript.

The general mathematical formulation of the detectors needs a bit of explanation. To take eq (5) as an example, it is written as Λ = (equation) > T. I think we should interpret this as meaning Λ is defined as (equation), and if Λ > T, an iceberg has been detected – This is absolutely correct. We have updated the text to make this more clear.

Detailed comments (only go as far as section 4)

11 ‘small sized’ - say what this means – thank you, we have elaborated on this.

15 (and many times subsequently) ‘pol’ → ‘polarisation ­Thank you for your suggestion, this has been changed.

16 ‘naught’ → ‘nought’ - Thank you for your suggestion, this has been changed.

18 ‘by reducing the channels to HH and VV’ this is not intelligible – Thank you for pointing this out. This has been changed to ‘using only the channels HH and VV’. We hope this is clearer.

18/19 ‘by using intensity channels.. T22’ again – the reader can not understand what this means – Thank you for pointing this out. This has been changed to ‘by using single intensity channels’ and we have removed the channel names to be explained in later sections.

38 ‘Smaller icebergs are larger hazards...’ How small? Does this inverse relationship between size and hazard extend to the smallest icebergs? Please give a reference. – We have defined small icebergs to be less than 120 m on the horizontal axis, and have updated this section to explain that large numbers of icebergs are produced from Greenland glaciers, we also added a reference.

44 ‘polarimetric behaviour of targets’ Well, I suppose I know what you mean by this phrase but it’s not really precise. – Thank you for your suggestion. We have updated this to refer to the structure of the scattering and covariance matrices.

53 ‘appear brighter’ → ‘have higher backscatter’ - Thank you for your suggestion, this has been changed.

54 ‘Previous work found...’ The connection of these two sentences to the previous text is not explained. – Thank you for pointing this out. We have updated this saying ‘Additionally’. We hope the connection is now clearer.

56 ‘...and then the sea clutter. - or, I assume, in the opposite order? – Thank you for this. We have added a sentence to explain both reflections happen at the same time.

63 ‘This is likely due to… toppling over’ I don’t understand why this would be true. – It is known that the dielectric constant properties of an ice body will increase with the presence of surface liquid water. Since these surfaces are smooth it means that the backscattering of surfaces will generally become more directive and therefore decrease in the back direction. This is explained in additional sentences added to the paper.

65 ‘pixel sizes are smaller (1-4 pixels)’ Smaller than what? And what does 1-4 pixels mean? Sorry about this, we removed this unfortunate expression.

68 ‘higher’ → ‘larger’ - Thank you for your suggestion, this has been changed.

73 ‘on ALOS...’ → ‘applied to ALOS...’? - Thank you for your suggestion, this has been changed.

81 ‘with PolSAR for icebergs in L band’ → ‘applied to L-band PolSAR data’ (perhaps – the meaning isn’t clear) - Thank you for your suggestion, this has been changed and we hope the meaning is a bit clearer.

81 ‘PolSAR’ - is this a typo for PALSAR? Or an unexplained abbreviation for ‘polarimetric SAR’? - Thank you for this, this was an abbreviation, this has been changed to reflect this.

92 ‘Where’ doesn’t begin a new sentence, so don’t capitalise it. (Happens elsewhere too.) - Thank you for your suggestion, this has been changed.

92 ‘… H is a linear horizontal wave’ This paragraph needs to be worded more carefully. The reader doesn’t know whether HV etc represents an amplitude, intensity, backscatter coefficient, or what. – Thank you for pointing this out. We have updated this section as follows. “where H is a linear horizontal wave and V is a linear vertical wave. The first letter indicates the transmitted wave and the second indicates the received wave. The elements of the matrix are the complex backscattering and they are called polarisation channels. HH and VV are known as co-channels and VH and HV are known as cross-channels."

95 ‘deterministic polarimetric target’ - what is this? – Thank you. Deterministic targets are fully polarised targets. We have added a definition here as  ‘A single target is any deterministic (fully polarised) polarimetric target that does not change spatially or temporally

102 ‘three-dimensional’ - but in eq. (2) you show it as four-dimensional. – Thank you. We show it as four dimensional for a general basis, but it is three dimensional with a monostatic sensor and reciprocal medium

104 ‘on the ground’ → ‘on the ocean surface’?? - Thank you for your suggestion, this has been changed.

105 ‘nondeterministic’ - what does this mean here?  - We mean that it is the opposite of deterministic, in which a target has a random scattering and therefore is partially polarised. In the text we added, “The target has a random scattering so that the polarimetric behaviour changes spatially and temporally

108 ‘partial target’ - what is this? – We have updated this to mean any non-deterministic target that has a polarimetric behaviour, which varies spatially and temporally. We have added it in the text:  “The targets on the ground are often distributed and non-deterministic (i.e. the target has a random scattering so that the polarimetric behaviour changes spatially and temporally), and they are known as partial targets

110 you define the matrix C but not T. – T is the coherency matrix when we use the Pauli basis to represent the scattering vector. We have added it in the text. “We can extract the second order statistics of a partial target using a covariance matrix [C] or a coherency matrix [T] if we are expressing the target in terms of a Pauli vector:”

113 ‘averaging’ - over what? – By this, we mean spatial averaging over the neighbouring pixels. We have added it in the text: “where ⟨. ⟩ is an averaging operator (i.e. spatially averaging over neighbour pixels) and * refers to the complex conjugate”

117 what do the lower case ‘s’ refer to in eq. (4)? Are these the same terms as in eq (1) (where you used upper case?) - Thank you for pointing out this error with the casing, this has been changed as they are the same terms as eq (1).

131 ‘review the literature to which they have been applied’ This isn’t what you mean. Surely ‘review the literature that describes their application’, or something like that? - Thank you for your this, you are correct, this has been changed.

139 ‘HV and VH (or HH, VV)’ very confusing. We meant cross-pol or co-pol channels. We have added this to the text as “They are based on the intensity of the cross and co polarisation channels

140 ‘Due to volume scattering...’ this sentence belongs in the introduction, not here. - Thank you for this, this has been moved to the introduction

145 ‘sizes… are spatial[ly] averages’ This isn’t what you mean. Please reword it. - Thank you for pointing this error out, this has been changed.

145 ‘spatially’ → ‘spatial’ - Thank you for your suggestion, this has been changed.

152 ‘if there is a reduction in volume’ - of what? – Thank you. This refers to a target leaving the target area.

152 ‘The complex nature of icebergs...’ this belongs in the introduction. - Thank you for pointing out this, this has been moved to the introduction

155 ‘The detector increased the contrast...’ How is this defined? How is reduction in clutter defined? – This is defined as the intensity of the target over the intensity of the clutter. We added it in the text “The detector improved the contrast (the intensity of the target over the intensity of the clutter) between icebergs and sea ice clutter by up to 75 times, greatly increasing the probability of accurate detection

167 Presumably Λ and T here are not the same as in eq (5)? – Thank you for highlighting this. We have updated eq (5) with Λ_a and Λ_b in eq (6).

172 ‘on if’ → ‘on whether’, or just ‘whether’, would be more idiomatic here I think - Thank you for your suggestion, this has been changed into ‘on whether’

175-179 I don’t understand the relation between eqq (7) and (8) – This is explained in the Appendix of a paper by Marino detecting icebergs using the depolarisation ratio anomaly detector. It can be referred to here: 

Marino, A.; Dierking, W.; Wesche, C. A depolarization ratio anomaly detector to identify icebergs in sea ice using dual-polarization SAR images. IEEE Transactions on Geoscience and Remote Sensing 2016, 54, 5602-5615.

182 ‘the area between the testing...’ I think a diagram would help enormoulsy here! – Thank you for pointing this out. You can refer to a figure produced within the paper (Figure 7).

200 Why has the symbol for a detector been changed from lambda to gamma here? Does it signifiy something different? – Thank you. We preferred to use different symbols to ensure that the different detectors were easily differentiated. iDPolRAD and Notch are different to each other.

205 ‘As this algorithm is quite general’ In what way is it more general than the others? – We have updated this. In the text we say that, “As this algorithm is focused on targets different from the sea (not just ships), it can also be used for the detection of other polarimetric targets, such as icebergs.”

208 ‘For the sake of brevity’ ?? I don’t understand. You present another equation that differs from the previous one only by the substitution of one variable for another. - Thank you for pointing this out, we are aware of the multiple equations in the literature used for different applications so felt they did not need to all be included. We have removed this sentence.

220 contract’ → ‘contrast’? - Thank you for pointing out this small error, this has been changed.

225 Please explain eq (12) more. – Thank you, we have included a bit more information about eq (12) here. In the text we added, “Since the quadratic forms represent power of a specific projection vector, this detector finds the optimal projection vector in the space of unitary polarimetric targets that provides the maximum ratio between the power of target over clutter.”

248 What are X and Σ in eq (14)? – X refers to the scattering vectors, while sigma refers to the covariance matrix. The small letters t and c refer to target and clutter. We added the following in the text: “Here X and Xt are scattering vectors and Σt and Σc are covariance matrices for target and clutter. This function is a quadratic used to construct an image

267 What are r and Y? (I now see they are defined later. Please move the definitions nearer to the equations that use them). - Thank you for your structural suggestion, this has been changed.

268 is ‘tr’ here different from ‘Trace’ in eq (2)? – There is no difference in tr and Trace. We have changed eq (2) to account for this.

269 ‘It is suggested… known to obey’ If it is known to follow a gamma distribution, the suggestion is unnecessary. If it has to be suggested, it can’t be known. - Thank you for pointing this out, this has been changed.

288 ‘;’ → ‘,’ - Thank you for pointing out the punctuation error, this has been changed.

294 ‘three images… average incidence angle’ not clear what this means – what are you averaging over? – We are reporting the incidence angle in the middle of the image. We added this info in the text “All data are quad-polarimetric, with an ascending mode and Single Look Complex (SLC) format with three images having an average incidence angle of 39° (in the centre of each image) and one with 31°.”

296 ‘naught’ → ‘nought’ - Thank you for your suggestion, this has been changed.

297 ‘polarimetric behaviour of icebergs’ → ‘polarimetric behaviour : of radar scattering from icebergs’ - Thank you for your suggestion, I can see that this is more accurate, this has been changed.

303 Table 1. Why does each image have three incidence angles? – We are using the mid, near and far range of each image. This has been added to the table caption.

317 ‘on if’ → ‘on whether’ - Thank you for your suggestion, this has been changed.

319 ‘if it was suspected… not an iceberg’ How can you tell? – We know that ships present as elongated shaped targets on an image, so we know these are not icebergs.

322 ‘We present Blosseville Coast.. as Blosseville’ - I don’t understand this statement. - Thank you. In our previous paper, we used two images for Blosseville Coast, one for north, and south. We have included this additional information.

328 alpha, entropy haven’t been defined. How were these calculated? – Thank you for this. An explanation of how alpha and entropy is now provided. In the text we added: “The alpha and entropy are calculated using the Cloude-Pottier eigenvalue/eigenvector polarimetric decomposition [55]. In this case, the entropy is calculated as the logarithmic sum of the eigenvalues of the covariance matrix. It refers to the randomness of the scattering behaviour and can be used to estimate the depolarisation of icebergs. Alpha is calculated from the eigenvectors of the covariance matrix. It is related to the incidence angle and dielectric constant of the scatterers [56]. Therefore it can be used to determine a particular scattering mechanism (odd bounce, even bounce, dihedral scattering).

358 ‘we exclude smaller icebergs (<100 m).’ I don’t understand. If you know that they are icebergs, why exclude them? Maybe you mean that you couldn’t reliably detect icebergs smaller than 100 m? - Thank you for pointing this out, you are correct, we could not reliably detect icebergs smaller than 120 m, this has been changed. In the text we added  “It is important to note that due to missing in-situ validation data, this analysis is restricted to icebergs that can be identified through visual analysis, so we cannot reliably detect smaller icebergs (<120 m).

372 ‘yellow areas’ - Please identify – I can’t see any! – You are correct, this has been corrected to ‘green areas’.

373 ‘Note the transparency of the polygons does not affect the analysis.’ I don’t understand this remark.  – Thank you for pointing this out, it creates confusion so we have removed it.

378 ‘reducing the channels to HH and HV’ - I don’t understand what you mean. – Thank you. This has created confusion, so we have changed ‘reducing’ to ‘using’.

387 ‘an image with pixels’ - are there other kinds? – This has been changed to ‘image pixels’ instead.

388 ‘train window’ - please could you refer to it as a ‘training window’? ‘Train window’ conjures up entirely the wrong mental image! – We can see that reviewer 2 has commented on this and have gone with their suggestion to refer to training window as ‘clutter window’.

394 ‘, too’ → ‘too,’ - Thank you for this, this has been changed.

397 ‘Cloude-Pottier decomposition’ - I think it’s the first we’ve heard about this. Please include in methods – Thank you, you can now find a small section explaining this in the methods section.

407-420 ‘it is clear...tend to give visually...’ It’s odd that a paper which has given such a lot of mathematical detail earlier on should rely on quite qualitative assessments of performance. This remains an issue through the results section. – Thank you. We obtained a more rigorous quantitative analysis in the following section, but in this section we have a preliminary visual analysis.

Reviewer 2 Report

The reviewer would like to thank the authors for this manuscript. The manuscript is informative and intuitively developed with reference to concepts from the information theory recalled into the domain of polarimetric remote sensing. The reviewer would like to appreciate the simple mathematical treatment introduced by the authors. The reviewer would like to seek some clarifications and would advise the authors to follow the suggested changes in the manuscript for the benefit of the community. The authors are requested to revise the manuscript as per the comments below.

Line 9: 

The authors are requested to replace ‘during polar nights’ with ‘during day & night passes’.

Line 9: 

The authors are requested to remove this sentence since it does not contribute any important information.

Line 10:

It will be better if the authors call it ‘polarimetric target detectors’.

Line 11:

The terms ‘performance and ability’ sounds synonymous. Why are the authors only targeting small sized icebergs? Is there a standard metric to classify the size of the icebergs as ‘small’, ‘medium’, or ‘large’?

Line 12:

The authors are not required to define all the polarimetric target detectors in the abstract.

Line 15:

It is not clear what the authors mean by ‘the data were calibrated and processed using sigma naught’? Moreover, these are methodical details that should not be ideally presented in the abstract. The authors are requested to remove such sentences (Line 16-20). Abstract should be brief and only convey the contributions unique to the paper.

Line 22:

The information presented in this sentence is not clear. It should be clear to the readers if the accuracies are expressed in percentage. Is it ‘false alarm’ or ‘false alarm rate’? It is not clear what the authors mean by ‘accuracies are 0.99 in open ocean and 0.90 in sea ice’.

Line 23-29:

What do the authors mean by ‘detectors show an overall reduction of performance’? What do the authors mean by ‘bigger icebergs’? Moreover, this contradicts the previously stated objective of this work to detect ‘small sized icebergs’. Are icebergs classified by their length? In summary, the authors are requested to rewrite the abstract and break the information into a few smaller sentences. At the moment the abstract does not convey a clear overall idea about the work. 

Line 33-40:

The authors state several points surrounding the Arctic. The authors are requested to cite relevant references for each one of these points.

Line 40-41:

What does the authors mean by SAR’s ability to function in ‘remote environments’? Is it not an obvious feature of just any satellite observation?

Line 51-57:

The authors are requested to not deviate the attention of the readers towards ship detection. Line 52-57 can be omitted. This also applies to Line 59-60.

Line 58:

By ‘clean sea’ the reviewer understands that the author means ‘calm sea’. The authors are requested to clarify this and reconsider using the suggested expression for better clarity.

Line 61-64:

The authors are requested to cite the following articles reporting the polarimetric scattering behaviour.

-Jagdhuber et al., Remote Sensing, 2014. “Identification of soil freezing and thawing states using SAR polarimetry at C-band”.

-Park, Remote Sensing, 2015. “Variations of microwave scattering by seasonal freeze/thaw transition in the permafrost active layer observed by ALOS PALSAR polarimetric data”.

-Muhuri et al., IEEE JSTARS, 2017. “Scattering mechanism based snow cover mapping using Radarsat-2 C-band polarimetric SAR data.” 

-Touzi, IEEE TGRS, 2007. “Target scattering decomposition in terms of roll-invariant target parameters.”

Furthermore, the authors should mention that such scattering is not limited to ‘irregular shaped icebergs toppling over’ but even the ice volume characteristics.

Line 64-66:

This sentence does not convey a clear idea. What do the authors mean by ‘pixel sizes are smaller’ and ‘multi-analysis is required in different window sizes’?

Line 66-67:

Please read this sentence again. ‘Does not negate the inability of detectors to identify...’. Please rephrase since the idea is not clear.

Line 67-72:

As a 3D body it is not quite clear which part of the iceberg is 120 m. ‘Smaller mass allows a greater potential to drift into the shipping lanes’, the authors are requested to cite a reference supporting this statement. Moreover, is it always true since large bodies can be driven by the additional force of wind.

Line 73-74:

The authors are requested to justify the selection of the satellite data for this kind of investigation. 

Line 74-75:

‘A combination of scattering mechanisms will affect the detection performance’. The authors are requested to reconsider this sentence if it is necessary to mention this information in this paper. If necessary, the authors should justify the role of scattering diversity in impacting the performance of detection.

Line 76-78:

This sentence does not convey any useful information. It is quite obvious in any paper to refer to the previous investigations.

Line 85-118:

The reviewer feels the information provided in this part of the manuscript is quite well known and the authors can directly start with discussing the polarimetric detectors. Moreover, the information provided in Line 119-128 can be integrated while discussing the detectors at a relevant location.

Line 146:

The reviewer would like to know if the terms ‘training’ and ‘testing’ can be renamed. These words are a bit misleading since there is no actual ‘training’ and ‘testing’ involved as in case of typical learning algorithms. How about ‘target and clutter window’?

Line 149:

The authors are requested to explain what they mean by ‘detection is triggered if an iceberg of the right size is found’. How is the size of the training and testing window determined with regards to the size of the iceberg in question? How is the relationship established between the size of the iceberg (say 120 m as mentioned by the authors) and the detection threshold?

Line 151-152:

The authors don’t provide sufficient background for this sentence. What do the authors mean by ‘surface is homogeneous’, ‘detector is equal to zeros’, and ‘detector becomes negative’? What do the authors mean by ‘reduction in volume’? The volume of ice or the percentage of volume scattering over a pixel.

Line 193-194:

The previous detectors are also in a manner based on the difference in the polarimetric properties. Furthermore, the authors are requested to leave the topic of ship detection and talk only about the applicability of the detectors for iceberg detection in terms of the polarimetric properties.

Line 297:

The authors are requested to clearly state at just one location the focus of this work. Moreover, it is not necessary to state the objectives of the previous investigations unless it is continuation of the same work/project.

Line 316-318:

It is not clear what the authors would like to convey here.

Line 631:

What do the authors mean by ‘easier scenarios’?

Line 662:

‘The quad-pol detectors provide the best detection’. Is this not an expected outcome?

Author Response

The reviewer would like to thank the authors for this manuscript. The manuscript is informative and intuitively developed with reference to concepts from the information theory recalled into the domain of polarimetric remote sensing. The reviewer would like to appreciate the simple mathematical treatment introduced by the authors. The reviewer would like to seek some clarifications and would advise the authors to follow the suggested changes in the manuscript for the benefit of the community. The authors are requested to revise the manuscript as per the comments below.

Line 9:  The authors are requested to replace ‘during polar nights’ with ‘during day & night passes’. – Thank you for your suggestion, this has been changed.

Line 9:  The authors are requested to remove this sentence since it does not contribute any important information. - Thank you for your suggestion, this has been removed

Line 10: It will be better if the authors call it ‘polarimetric target detectors’. -  Thank you for your suggestion, this has been changed.

Line 11: The terms ‘performance and ability’ sounds synonymous. Why are the authors only targeting small sized icebergs? Is there a standard metric to classify the size of the icebergs as ‘small’, ‘medium’, or ‘large’? – Thank you. We are targeting small sized icebergs because they are much harder to detect. The classification table for iceberg size is available in the literature and can be found here:

Jackson, C.R.; Apel, J.R. Synthetic Aperture Radar: Marine User’s Manual; US Department of Commerce, National Oceanic and Atmospheric Administration, National Environmental Satellite, Data, and Information Service (NESDIS), Office of Research and Applications: Washington, DC, USA, 2004; p. 411.

Line 12: The authors are not required to define all the polarimetric target detectors in the abstract. - Thank you for your suggestion, this has been changed.

Line 15: It is not clear what the authors mean by ‘the data were calibrated and processed using sigma naught’? Moreover, these are methodical details that should not be ideally presented in the abstract. The authors are requested to remove such sentences (Line 16-20). Abstract should be brief and only convey the contributions unique to the paper. - Thank you for your suggestion, these sentences have been removed.

Line 22: The information presented in this sentence is not clear. It should be clear to the readers if the accuracies are expressed in percentage. Is it ‘false alarm’ or ‘false alarm rate’? It is not clear what the authors mean by ‘accuracies are 0.99 in open ocean and 0.90 in sea ice’. - Thank you for your suggestion, in this case, it is false alarm rate, however I can see that the term ‘accuracy’ may be confusing, and we use it to define the ‘probability of detection’ at a fixed false alarm. For clarity we changed this in the abstract.

Line 23-29: What do the authors mean by ‘detectors show an overall reduction of performance’? – An overall reduction of performance refers to the ROC curves for dual and single pol images getting lower.

What do the authors mean by ‘bigger icebergs’?  - Here, we mean icebergs have brighter backscattering signals and are occupying more pixels.

Moreover, this contradicts the previously stated objective of this work to detect ‘small sized icebergs’. – Although we focus on smaller icebergs, their size is still variable and larger ones a few hundred metres in size will still be analysed here.

Are icebergs classified by their length? – They are classified by their length as shown in Jackson and Apel’s paper which all literature refers to. By length we mean the horizontal axis of the iceberg.

In summary, the authors are requested to rewrite the abstract and break the information into a few smaller sentences. At the moment the abstract does not convey a clear overall idea about the work.

Thanks for your suggestion, we made several changes to the Abstract and we hope it is clearer.

Line 33-40: The authors state several points surrounding the Arctic. The authors are requested to cite relevant references for each one of these points. – Thank you for your request. Some references are now cited for each of these points.

Line 40-41: What does the authors mean by SAR’s ability to function in ‘remote environments’? Is it not an obvious feature of just any satellite observation? – Thank you for this. We mean harsh environments with no solar illumination during polar nights such as in Greenland. We changed the text as follows. “Synthetic Aperture Radar (SAR) is well known for the ability to function in harsh environments with no solar illumination and it is especially useful in areas with extended cloud cover or polar nights

Line 51-57: The authors are requested to not deviate the attention of the readers towards ship detection. Line 52-57 can be omitted. This also applies to Line 59-60. - Thank you for your suggestion, we have updated this section to reflect on this, but we are also aware that a lot of iceberg detection relies on algorithms applied to ship detection previously.

Line 58: By ‘clean sea’ the reviewer understands that the author means ‘calm sea’. The authors are requested to clarify this and reconsider using the suggested expression for better clarity - Thank you for your suggestion, we have changed this to increase the clarity.

Line 61-64: The authors are requested to cite the following articles reporting the polarimetric scattering behaviour.

-Jagdhuber et al., Remote Sensing, 2014. “Identification of soil freezing and thawing states using SAR polarimetry at C-band”.

-Park, Remote Sensing, 2015. “Variations of microwave scattering by seasonal freeze/thaw transition in the permafrost active layer observed by ALOS PALSAR polarimetric data”.

-Muhuri et al., IEEE JSTARS, 2017. “Scattering mechanism based snow cover mapping using Radarsat-2 C-band polarimetric SAR data.”

-Touzi, IEEE TGRS, 2007. “Target scattering decomposition in terms of roll-invariant target parameters.”

Furthermore, the authors should mention that such scattering is not limited to ‘irregular shaped icebergs toppling over’ but even the ice volume characteristics. – Thank you for all these references, these have been cited and added as well as the additional information regarding ice volume characteristics. Indeed a wetter surface reduces the penetration and therefore the volume scattering. We reported on this in a previous paper too.

Line 64-66: This sentence does not convey a clear idea. What do the authors mean by ‘pixel sizes are smaller’ and ‘multi-analysis is required in different window sizes’? – Thank you for pointing this out. We have changed this line to state that the scattering behaviour of a target can be affected by pixel size.

Line 66-67: Please read this sentence again. ‘Does not negate the inability of detectors to identify...’. Please rephrase since the idea is not clear. – Thank you for pointing this out. We have rephrased the line here as. “However, detectors can still identify icebergs at sizes of roughly 1-4 pixels.”

Line 67-72: As a 3D body it is not quite clear which part of the iceberg is 120 m. ‘Smaller mass allows a greater potential to drift into the shipping lanes’, the authors are requested to cite a reference supporting this statement. – Thank you for pointing this out. The horizontal axis of an iceberg is used in iceberg size classification. This the horizontal size above the water surface. We have changed the sentence requiring a citation to state instead that icebergs can go undetected due to oceanographic and meteorological conditions.

Moreover, is it always true since large bodies can be driven by the additional force of wind.

Line 73-74: The authors are requested to justify the selection of the satellite data for this kind of investigation. – Thank you for the request. ALOS-2 quad-pol data have a high spatial resolution, and L-band can penetrate ice bodies, which can show more volume scattering

Line 74-75: ‘A combination of scattering mechanisms will affect the detection performance’. The authors are requested to reconsider this sentence if it is necessary to mention this information in this paper. If necessary, the authors should justify the role of scattering diversity in impacting the performance of detection. – Thank you for the request. We have removed this sentence.

Line 76-78: This sentence does not convey any useful information. It is quite obvious in any paper to refer to the previous investigations. – Thank you for pointing this out. We have changed this sentence to reflect on previous detection methods.

Line 85-118: The reviewer feels the information provided in this part of the manuscript is quite well known and the authors can directly start with discussing the polarimetric detectors. Moreover, the information provided in Line 119-128 can be integrated while discussing the detectors at a relevant location. – Thank you for suggesting this, Indeed the first 4 paragraphs of the polarimetry section are well known but it helps us in defining all the symbols used in the following sections, and a reader without background in polarimetric SAR may feel lost. This is why we would humbly prefer to keep these paragraphs.

Line 146: The reviewer would like to know if the terms ‘training’ and ‘testing’ can be renamed. These words are a bit misleading since there is no actual ‘training’ and ‘testing’ involved as in case of typical learning algorithms. How about ‘target and clutter window’? – We have changed the names to target and clutter window thought the entire manuscript.

Line 149: The authors are requested to explain what they mean by ‘detection is triggered if an iceberg of the right size is found’. How is the size of the training and testing window determined with regards to the size of the iceberg in question? – Here, we mean that the size of the clutter and target windows determine the size of the detected iceberg in question.

How is the relationship established between the size of the iceberg (say 120 m as mentioned by the authors) and the detection threshold.

For the iDPolRAD the threshold is set using a CA-CFAR since the statistics of the iDPolRAD resample an exponential. We did not do any study yet which links the size to the false alarm rate (FAR) to use for detecting the iceberg. 

Line 151-152: The authors don’t provide sufficient background for this sentence. – Here we have addressed each comment separately.

What do the authors mean by ‘surface is homogeneous’

The clutter is spatially homogeneous, we added the word “spatially” in the text

, ‘detector is equal to zeros’

The value of the detector equals zero

, and ‘detector becomes negative’

The values of the detector become a negative number? We changed this in the text.

What do the authors mean by ‘reduction in volume’? The volume of ice or the percentage of volume scattering over a pixel.

The volume scattering amount reduces from the clutter to the target window. We added this in the text.

Here the text is as follows: “If the surface clutter is spatially homogeneous, then the numeric value of the detector is equal to zero and if there is a reduction in volume (the volume scattering reduces from the clutter to target window), the numeric value of the detector becomes negative.”

Line 193-194: The previous detectors are also in a manner based on the difference in the polarimetric properties. Furthermore, the authors are requested to leave the topic of ship detection and talk only about the applicability of the detectors for iceberg detection in terms of the polarimetric properties. – Thank you for the request. We have changed the word ship to ‘targets at sea’.

Line 297: The authors are requested to clearly state at just one location the focus of this work. Moreover, it is not necessary to state the objectives of the previous investigations unless it is continuation of the same work/project. – Thank you for this suggestion. In order to make the analysis more general, we used more than one location. This work is built upon the results of the first paper where the physics was analysed. Here we want to evaluate the detection ability.

Line 316-318: It is not clear what the authors would like to convey here. – Here, we are looking to provide some information on how the visual analysis was done to identify icebergs, such as how we distinguished them from ships.

Line 631: What do the authors mean by ‘easier scenarios’? – We have changed this to ‘detection task’ instead of scenario.

Line 662: ‘The quad-pol detectors provide the best detection’. Is this not an expected outcome? – Yes, we agree with the reviewer. We did expect that quad-pol would perform better, but in the community, some scientists still do not believe this is the case, so we have included this here.

Round 2

Reviewer 1 Report

This reads more clearly than v1, and the presentation of results and their discussion is generally good. The introduction reads quite like a textbook on radar polarimetry and I'm not clear how much of this mathematical development is actally needed for what follows. Some of the notation doesn't seems to be used, and some notation and indeed concepts used in the experimental part of the work are not explained in the introduction.

In section 2, please straighten out the mathematical formulation of the detectors. For example, (eq. 5) expresses two relationships: an equality and an inequality. The equality is the definition of the term lambda_a, while the inequality is (I must assume) the test applied by the detector. In other words, the first part of this equation is always true, while the second part may or may not be true. Eq 7 is harder still to interpret since it contains two equalities and an inequality. This is a specialised use of mathematical notation so you should explain it. (Use of ≡ instead of = in appropriate places would certainly help). Reflection symmetry XC (eq 13) does not seem to be formulated the same way (not an inequality).

Proofreading is still quite poor and I picked up a lot of errors or poorly phrased statements.

(42) ‘have also...’ implies additionally to something else – what are the other kinds of detectors?

(53) “clean sea’ - perhaps you could explain what you mean by this phrase? It could easily be read in the wrong way.

(57) omit second comma

(58) ‘irregular’ → ‘irregularly’

(59) ‘explained in [23]’ → ‘explained by Bailey et al. [23]’. (The idea is that you can read the sentence out loud with the bracketed reference omitted and it still makes sense).

(79) ‘therefore has a higher spatial resolution’ - higher than what? And why does this higher spatial resolution follow from the frequency and/or polarisation characteristics?

(79) ‘23 m’ → 23 cm

(85) ‘cross-polarisation’ add ‘backscaterring coefficient’ after these words

(93 and subsequently) This may just be a matter of taste, but I don’t much like the abbreviation of ‘polarisation’ to ‘pol’.

(103) ‘They’ → ‘It’

(108) ‘all four diagonal elements’ - surely two?

(110) ‘three-dimensional’. The definition in (2) is 4-dimensional. Explain how it reduces to a three-dimensional form.

(128) ‘icebergs are generally dominated by...’ this is a poor construction but I suppose its meaning is clear enough

(133) ‘backed up’ - rather informal. ‘supported’ would be better.

(143) ‘only with ships’ - I think you mean ‘only with radar imagery of ships’ or something like that

(153) ‘The two window sizes… spatial averages’ This clause needs rephrasing – it is not the sizes that are averages.

(158) The size of the target and clutter windows...’ I don’t think this sentence means what it says, though I’m not quite sure what it does mean. Perhaps that the window sizes determine the size of the detectable icebergs?

(165) ‘The detector improved the contrast...’ How is this possible? The output of the detector is not an intensity but a binary variable. Similar comment for the clutter – how does the algorithm reduce it?

(201) ‘case study… Sentinel-1’ Not clear if the case study is part of the present work, but if it is you should explain why it is relevant to use C band radar data since the focus of the present work is L band.

(206) ‘proposed in [36]’ see comment on line (59)

(208) why ‘final’?

(212) Not clear how t relates to the general formulation in section 1.1.

(217) ‘polarimetric targets’ → ‘polarised targets’?

(219) again, why is this ‘final’?

(231) Not clear how this relates to the general polarimetric formulation in section 1.1. How is omega defined? What is c?(Is it the same as the covariance matrix??) What are sigma eigenvectors of?

(241) ‘12’ should be subscripted.

(257) Scattering vectors were defined in (2) and (4) but given the symbol k. Is this a different kind of scattering vector, or just a change of notation? Similarly, covariance matrix was C before.

(259) ‘covariance matrix of… iceberg’ rather loose wording. It’s the covariance matric of the radar backscatter from the iceberg.

(281) ‘y is the speckle’ - how is this defined?

(285) ‘obey’ → ‘obeys’

(287) ‘textured scenario’ please reword this

(321) Give unit of resolution (m) in table heading

(336) ‘if’ → ‘whether’

(342) ‘depolarisation of icebergs’ - again, rather loose wording

(405) ‘Dual pol detection...’ sentence is somewhat unclear

(407) ‘removing the detectors’ - what does this mean?

(420) ‘detection images...’ Now I am confused. The detectors defined in section 2 have (as I understand it) binary output, but here we have images representing continuous variables. How are they defined?

(429, 432) ‘Colourmaps’ → ‘greyscales’?

(448) ‘plot’ → ‘plots’

(498) ‘This is because’ - what? Sentence is incomplete.

(574) ‘-5’ → 10^-5.

(579) ‘detectors set… mechanism’ - I don’t understand what this means

(623) ‘highest… values’ There is more than one ‘highest value’ per table, so I am not sure I know how to interpret this statement.

(650) omit semicolon (or replace with comma)

(660) ‘we are open ocean’ - what?

(661) ‘0.99 of quad’ → ‘0.99 for quad’

(665) same as (661)

(667) ‘similar’ → ‘similarly’

(690) ‘3242 icebergs’ - was this fact mentioned before?

(691) ‘that overall;’ → ‘that, overall,’

(692) lambda → sigma?

Author Response

This reads more clearly than v1, and the presentation of results and their discussion is generally good. The introduction reads quite like a textbook on radar polarimetry and I'm not clear how much of this mathematical development is actally needed for what follows. Some of the notation doesn't seems to be used, and some notation and indeed concepts used in the experimental part of the work are not explained in the introduction.

In section 2, please straighten out the mathematical formulation of the detectors. For example, (eq. 5) expresses two relationships: an equality and an inequality. The equality is the definition of the term lambda_a, while the inequality is (I must assume) the test applied by the detector. In other words, the first part of this equation is always true, while the second part may or may not be true. Eq 7 is harder still to interpret since it contains two equalities and an inequality. This is a specialised use of mathematical notation so you should explain it. (Use of ≡ instead of = in appropriate places would certainly help). Reflection symmetry XC (eq 13) does not seem to be formulated the same way (not an inequality). - Thank you for the comments. Here, the inequality is used to symbolise that the detector is obtained by setting a threshold and that anything not detected is below the threshold while anything detected is above the threshold. It is a formalism that is commonly used in detection theory. The expression all in one line is indeed confusing so we have modified this to ensure it is clearer. The first line is the expression of the iDPolRAD operator, while the second line is the expression for the detector. For the reflection symmetry we have also made this change.

Proofreading is still quite poor and I picked up a lot of errors or poorly phrased statements.

(42) ‘have also...’ implies additionally to something else – what are the other kinds of detectors? – Thank you for pointing this out. We have now specified the two types of detectors in the text.

(53) “clean sea’ - perhaps you could explain what you mean by this phrase? It could easily be read in the wrong way. – Thank you. We added a sentence to specify that open water without wind effects can produce specular reflection.

(57) omit second comma – thank you, this has been omitted.

(58) ‘irregular’ → ‘irregularly’ – thank you, this has been revised.

(59) ‘explained in [23]’ → ‘explained by Bailey et al. [23]’. (The idea is that you can read the sentence out loud with the bracketed reference omitted and it still makes sense). – Thank you for pointing this out, we changed this accordingly.

(79) ‘therefore has a higher spatial resolution’ - higher than what? And why does this higher spatial resolution follow from the frequency and/or polarisation characteristics? – Thank you. We have changed this to ‘high spatial resolution’ instead as it creates confusion.

(79) ‘23 m’ → 23 cm – Thank you, this has been revised.

(85) ‘cross-polarisation’ add ‘backscaterring coefficient’ after these words – thank you, we added this in the text.

(93 and subsequently) This may just be a matter of taste, but I don’t much like the abbreviation of ‘polarisation’ to ‘pol’. – Thank you. We have revised the manuscript according to this, as we do not want to cause confusion.

(103) ‘They’ → ‘It’ – Thank you, this has been revised.

(108) ‘all four diagonal elements’ - surely two? – This is indeed correct, thank you for pointing this out. We have changed this.

(110) ‘three-dimensional’. The definition in (2) is 4-dimensional. Explain how it reduces to a three-dimensional form. – Thank you. The equation in (2) becomes three dimensional in the case of a monostatic sensor and reciprocal medium in which HV = VH.

(128) ‘icebergs are generally dominated by...’ this is a poor construction but I suppose its meaning is clear enough – Thank you, indeed this does not sound clear so we have changed it.

(133) ‘backed up’ - rather informal. ‘supported’ would be better. – thank you, we have changed this to ‘supported’.

(143) ‘only with ships’ - I think you mean ‘only with radar imagery of ships’ or something like that – thank you, yes we mean with SAR imagery of ships.

(153) ‘The two window sizes… spatial averages’ This clause needs rephrasing – it is not the sizes that are averages. – Thank you for pointing this out. You are correct, and we have rephrased this clause.

(158) The size of the target and clutter windows...’ I don’t think this sentence means what it says, though I’m not quite sure what it does mean. Perhaps that the window sizes determine the size of the detectable icebergs? - We are sorry for the confusion. The size of the detectable iceberg depends on the size of the target and clutter windows. We have modified this in the text.

(165) ‘The detector improved the contrast...’ How is this possible? The output of the detector is not an intensity but a binary variable. Similar comment for the clutter – how does the algorithm reduce it? – Thank you for the question. We first used the test statistic, which improves the contrast between the clutter and target. Then we apply the threshold. We added this into the text.

(201) ‘case study… Sentinel-1’ Not clear if the case study is part of the present work, but if it is you should explain why it is relevant to use C band radar data since the focus of the present work is L band. Thank you, we have removed this sentence as it does not have any relevance.

(206) ‘proposed in [36]’ see comment on line (59) – thank you we added the author name.

(208) why ‘final’? - Thank you, we removed the word final

(212) Not clear how t relates to the general formulation in section 1.1. -Here t is a vectorisation of the T matrix, so it has all the information content of T, but in a vector format rather than a matrix format. We have added the T matrix into section 1.1. 

(217) ‘polarimetric targets’ → ‘polarised targets’? – Thank you, this has been revised.

(219) again, why is this ‘final’? - Thank you. We have removed the word final

(231) Not clear how this relates to the general polarimetric formulation in section 1.1. How is omega defined? What is c?(Is it the same as the covariance matrix??) What are sigma eigenvectors of? - Thanks for the questions. More details are in the appendix of the journal paper. We agree that without following the derivation of the final formulas it is hard to see the link, but we would like to avoid including the 1 page appendix here because it will bring the attention away from the main subject of the paper. The parameters mentioned are defined in the text and alternatively we can remove the full session introducing the physical interpretation of the iDPolRAD. Omega are scattering mechanisms or projection vectors which are not scattering vectors.

(241) ‘12’ should be subscripted. – thank you, this has been changed.

(257) Scattering vectors were defined in (2) and (4) but given the symbol k. Is this a different kind of scattering vector, or just a change of notation? Similarly, covariance matrix was C before. - Thanks for pointing this out. As we have mentioned, we use C for a generic covariance matrix (it does not need to have a Pauli basis) and we use T for when it does need to have a Pauli basis. We agree that in eq (3) this should have been T rather than C so we have corrected this.

(259) ‘covariance matrix of… iceberg’ rather loose wording. It’s the covariance matric of the radar backscatter from the iceberg. – thank you for pointing this out, we changed this and hope it is clearer now.

(281) ‘y is the speckle’ - how is this defined? -Thank you for spotting this, we reworded this to “speckled image”

(285) ‘obey’ → ‘obeys’ – thank you this has been changed.

(287) ‘textured scenario’ please reword this - Thank you. We changed this into “scenario with texture”

(321) Give unit of resolution (m) in table heading – thank you, we added the unit to the table heading.

(336) ‘if’ → ‘whether’ – thank you, this has been revised.

(342) ‘depolarisation of icebergs’ - again, rather loose wording – Here, we mean that the entropy can be used to estimate whether polarimetric targets are depolarised, or have a more dominant scattering mechanism.

(405) ‘Dual pol detection...’ sentence is somewhat unclear – This has been changed to dual-polarimetric detection

(407) ‘removing the detectors’ - what does this mean? – We have removed this phrase as it causes confusion.

(420) ‘detection images...’ Now I am confused. The detectors defined in section 2 have (as I understand it) binary output, but here we have images representing continuous variables. How are they defined? - Good point, the detectors have thresholds, but the threshold is set on a distance generated by the detector algorithm. The distance is what we have shown on the images, since this allows to have a feel of the detection capabilities and the contrast (while a binary mask depends too strongly on the selection of threshold). In the text we called these images “detection distances.

(429, 432) ‘Colourmaps’ → ‘greyscales’? – Thank you, indeed these are black and white images so we have changed this.

(448) ‘plot’ → ‘plots’ – Thank you, we have revised this.

(498) ‘This is because’ - what? Sentence is incomplete. – thank you for pointing out, it turns out the rest of the sentence began after the figures so we have corrected this.

(574) ‘-5’ → 10^-5. – Thank you, we have changed this.

(579) ‘detectors set… mechanism’ - I don’t understand what this means – We have reworded this part of the sentence to ‘detectors set representing an appropriate mechanism’

(623) ‘highest… values’ There is more than one ‘highest value’ per table, so I am not sure I know how to interpret this statement. - Here we mean highest probability of detection value after fixing the Pf. So is the highest on the raw. We added this in the text.

(650) omit semicolon (or replace with comma) – Thank you, we have changed this.

(660) ‘we are open ocean’ - what? – Thank you for spotting this. We have reworded this section, “However, in open ocean, the detection task is easier”

(661) ‘0.99 of quad’ → ‘0.99 for quad’ – Thank you, we have updated this

(665) same as (661) – Thank you, same update applied.

(667) ‘similar’ → ‘similarly’ – Thank you, this has been revised.

(690) ‘3242 icebergs’ - was this fact mentioned before? – Thank you for spotting this, in total we did indeed select this number, we have added this in the text in section 4.2

(691) ‘that overall;’ → ‘that, overall,’ – Thank you, we changed this.

(692) lambda → sigma? - Thank you. Here lambda3 is the third eigenvalue of the Cloude-Pottier decomposition so we humbly believe it should remain as lambda here.